# Early life infection and proinflammatory, atherogenic metabolomic and lipidomic profiles in infancy: a population-based cohort study

Toby Mansell[1,2], Richard Saffery[1,2], Satvika Burugupalli[3], Anne-Louise Ponsonby[1,2,4], Mimi LK Tang[1,2,5], Martin O'Hely[1,6], Siroon Bekkering[1,7], Adam Alexander T Smith[3], Rebecca Rowland[1], Sarath Ranganathan[1,2,5], Peter D Sly[1,8], Peter Vuillermin[1,6,9], Fiona Collier[6,9], Peter Meikle[3†], David Burgner[2,10*†], Barwon Infant Study Investigator Group

[1]Murdoch Children's Research Institute, Parkville, Australia; [2]Department of Paediatrics, University of Melbourne, Parkville, Australia; [3]Metabolomics Laboratory, Baker Heart and Diabetes Institute, Melbourne, Australia; [4]The Florey Institute of Neuroscience and Mental Health, Parkville, Australia; [5]Royal Children's Hospital, Parkville, Australia; [6]Deakin University, Geelong, Australia; [7]Department of Internal Medicine, Radboud Institute for Molecular Life Sciences, Radboud University Medical Centre, Nijmegen, Netherlands; [8]Child Health Research Centre, University of Queensland, Brisbane, Australia; [9]Child Health Research Unit, Barwon Health, Geelong, Australia; [10]Department of Paediatrics, Monash University, Clayton, Australia

*For correspondence:
david.burgner@mcri.edu.au

†Joint senior authors

Group author details:
Barwon Infant Study Investigator Group See page 19

## Abstract

**Background:** The risk of adult onset cardiovascular and metabolic (cardiometabolic) disease accrues from early life. Infection is ubiquitous in infancy and induces inflammation, a key cardiometabolic risk factor, but the relationship between infection, inflammation, and metabolic profiles in early childhood remains unexplored. We investigated relationships between infection and plasma metabolomic and lipidomic profiles at age 6 and 12 months, and mediation of these associations by inflammation.

**Methods:** Matched infection, metabolomics, and lipidomics data were generated from 555 infants in a pre-birth longitudinal cohort. Infection data from birth to 12 months were parent-reported (total infections at age 1, 3, 6, 9, and 12 months), inflammation markers (high-sensitivity C-reactive protein [hsCRP]; glycoprotein acetyls [GlycA]) were quantified at 12 months. Metabolic profiles were 12-month plasma nuclear magnetic resonance metabolomics (228 metabolites) and liquid chromatography/mass spectrometry lipidomics (776 lipids). Associations were evaluated with multivariable linear regression models. In secondary analyses, corresponding inflammation and metabolic data from birth (serum) and 6-month (plasma) time points were used.

**Results:** At 12 months, more frequent infant infections were associated with adverse metabolomic (elevated inflammation markers, triglycerides and phenylalanine, and lower high-density lipoprotein [HDL] cholesterol and apolipoprotein A1) and lipidomic profiles (elevated phosphatidylethanolamines and lower trihexosylceramides, dehydrocholesteryl esters, and plasmalogens). Similar, more marked, profiles were observed with higher GlycA, but not hsCRP. GlycA mediated a substantial proportion of the relationship between infection and metabolome/lipidome, with hsCRP generally mediating a lower proportion. Analogous relationships were observed between infection and 6-month inflammation, HDL cholesterol, and apolipoprotein A1.

**Conclusions:** Infants with a greater infection burden in the first year of life had proinflammatory and proatherogenic plasma metabolomic/lipidomic profiles at 12 months of age that in adults are indicative of heightened risk of cardiovascular disease, obesity, and type 2 diabetes. These findings suggest potentially modifiable pathways linking early life infection and inflammation with subsequent cardiometabolic risk.

**Funding:** The establishment work and infrastructure for the BIS was provided by the Murdoch Children's Research Institute (MCRI), Deakin University, and Barwon Health. Subsequent funding was secured from National Health and Medical Research Council of Australia (NHMRC), The Shepherd Foundation, The Jack Brockhoff Foundation, the Scobie & Claire McKinnon Trust, the Shane O'Brien Memorial Asthma Foundation, the Our Women's Our Children's Fund Raising Committee Barwon Health, the Rotary Club of Geelong, the Minderoo Foundation, the Ilhan Food Allergy Foundation, GMHBA, Vanguard Investments Australia Ltd, and the Percy Baxter Charitable Trust, Perpetual Trustees. In-kind support was provided by the Cotton On Foundation and CreativeForce. The study sponsors were not involved in the collection, analysis, and interpretation of data; writing of the report; or the decision to submit the report for publication. Research at MCRI is supported by the Victorian Government's Operational Infrastructure Support Program. This work was also supported by NHMRC Senior Research Fellowships to ALP (1008396); DB (1064629); and RS (1045161) , NHMRC Investigator Grants to ALP (1110200) and DB (1175744), NHMRC-A*STAR project grant (1149047). TM is supported by an MCRI ECR Fellowship. SB is supported by the Dutch Research Council (452173113).

## Editor's evaluation

This paper provides data from a population-based cohort study on early life infection and proinflammatory, atherogenic metabolomic and lipidomic profiles at 12 months of age. The authors generated matched infection, metabolomics and lipidomics data from 555 infants in a pre-birth longitudinal cohort and they showed that frequent infant infections are associated with adverse metabolomic and lipidomic profiles. They also report that similar profiles are noted with higher glycoprotein acetyls (GlycA), but not hsCRP. The paper is interesting and assesses the role of infection and markers of inflammation on lipid and metabolic profile of patients. It provides a comprehensive analysis of lipids and metabolites in infants in response to infection.

## Introduction

Infectious diseases are ubiquitous in infancy and childhood, with potential long-term impacts on health across the life course. Infection has been recognised as a potential contributor to atherosclerotic cardiovascular disease (CVD), one of the leading causes of adult morbidity and mortality, since the 19th century (*Nieto, 1998*). More recent adult studies link previous infection with long-term risks of disease (*Bergh et al., 2017*; *Cowan et al., 2018*; *Wang et al., 2017*). The mechanisms are largely unknown, but likely include immune activation and heightened inflammation (*Shah, 2019*), which are pathways central to CVD pathogenesis (*Donath et al., 2019b*; *Ferrucci and Fabbri, 2018*) and therefore offer potentially druggable targets in high-risk individuals (*Donath et al., 2019a*; *Ridker et al., 2017*). High-sensitivity C-reactive protein (hsCRP) has been extensively used as a marker of chronic inflammation in adult studies but is an acute phase reactant in children and may not reflect chronic inflammation in early life. Glycoprotein acetyls (GlycA) is a nuclear magnetic resonance (NMR) composite measure that is suggested to better reflect cumulative, chronic inflammation (*Connelly et al., 2017*). GlycA is an emerging biomarker for cardiometabolic risk (*Connelly et al., 2017*) that outperforms hsCRP as a predictor of CVD events and mortality (*Akinkuolie et al., 2014*; *Duprez et al., 2016*), and of infection-related morbidity and mortality (*Ritchie et al., 2015*). For example, in the Multi-Ethnic Study of Atherosclerosis (n = 6523), higher GlycA was associated with increased risk of incidence CVD and death, even after adjustment for hsCRP and other inflammatory markers. Conversely, prediction of these outcomes by hsCRP attenuated to null after mutual adjustment (*Duprez et al., 2016*).

Cardiovascular and metabolic (cardiometabolic) disease pathogenesis begins in early life and accrues across the life course (*Nakashima et al., 2008*). Infections occur disproportionally in early

childhood (*Cromer et al., 2014*; *Grüber et al., 2008*; *Troeger et al., 2018*; *Tsagarakis et al., 2018*), and there is a dose-response relationship between childhood infections, adverse cardiometabolic phenotypes (*Burgner et al., 2015c*), and CVD events (*Burgner et al., 2015a*) in adulthood. Infection is linked to proatherogenic metabolic perturbations in later childhood and adulthood (*Feingold and Grunfeld, 2019*; *Khovidhunkit et al., 2000*), including higher triglycerides and oxidised low-density lipoprotein (LDL), and lower high-density lipoprotein (HDL) cholesterol and apolipoprotein A1 (ApoA1) (*Liuba et al., 2003*; *Pesonen et al., 1993*), and to acute and chronic inflammation (*Burgner et al., 2015b*; *Ritchie et al., 2015*), but little is known about these relationships in early life, when most infections occur.

We therefore aimed to characterise metabolomic and lipidomic profiles at 6 and 12 months of age and their relationship to infection burden during the first year of life. We also investigated the extent to which inflammation mediated the relationship between infection burden and metabolomic and lipidomic differences.

## Materials and methods

### Study cohort

This study used available data from 555 mother-infant dyads in the Barwon Infant Study (BIS), a population-based pre-birth longitudinal cohort (n = 1074 mother-infant dyads). The cohort details and inclusion/exclusion criteria have been detailed elsewhere (*Vuillermin et al., 2015*); in brief, mothers were eligible if they were residents of the Barwon region in south-east Australia and planned to give birth at the local public or private hospital. Mothers were recruited at approximately 15 weeks' gestation and provided informed consent. They were excluded if they were not a permanent Australian resident, aged <18 years, required an interpreter to complete questionnaires, or had previously participated in BIS. Infants were excluded if they were very preterm (<32 completed weeks gestation) or had a serious illness or major congenital malformation identified during the first few days of life. Ethics approval was granted by the Barwon Health Human Research Ethics Committee (HREC 10/24).

### Parent-reported infections

At the 4-week, 3-month, 6-month, 9-month, and 12-month time points following birth, mothers were asked to report each episode of infant illness or infection since the previous time point using standardised online questionnaires. The number of parent-reported infections from birth to 12 months was defined as the total number of respiratory tract infections, gastroenteritis, conjunctivitis, and acute otitis media episodes. In secondary analyses, numbers of parent-reported infections from birth to 6 months and from 6 to 12 months were considered. It was not possible to identify the proportion of parent-reported infections that lead to health service utilisation (*Rowland et al., 2021*).

### Other maternal and infant measures

Questionnaires during pregnancy and at birth were used to collect self-reported data on maternal age, household income, maternal education, and prenatal smoking (considered here as a dichotomous any/none exposure). Residential postcode was used to determine neighbourhood disadvantage using the Index of Relative Socio-Economic Disadvantage (IRSD) from the 2011 Socio-Economic Indexes for Areas (SEIFA) (*Pink, 2013*), with a lower score corresponding to greater socioeconomic disadvantage. Pre-eclampsia (based on International Association of Diabetes and Pregnancy Study Groups criteria; *Tranquilli et al., 2014*) and gestational diabetes (based on International Society for the Study of Hypertension in Pregnancy criteria; *Nankervis et al., 2013*) diagnoses were extracted from hospital records. Infant gestational age, birth weight, and mode of delivery (categorised as vaginal, planned caesarean section, or unplanned caesarean section delivery) were collected from birth records, and the age- and sex-standardised birth weight z-score was calculated using the 2009 revised British United Kingdom World Health Organisation (UK-WHO) growth charts (*Cole et al., 2011*). Postnatal smoking data was collected from questionnaire data, with mothers asked the average number of hours each day someone smoked near or in the same room as the child (*Gray et al., 2019*). This was dichotomised as any postnatal smoke exposure if >0 hr reported at any time point up to 12 months of age, or no postnatal smoke exposure. Breastfeeding duration up to 12 months of age was collected from maternal questionnaire data. As most evidence for the protective effect of breastfeeding on

early life infection is from comparisons between any breastfeeding and no breastfeeding (*Victora et al., 2016*), and in light of previous evidence in BIS for an association between even a short duration of breastfeeding and lower odds of infection in early infancy (*Rowland et al., 2021*), we first looked at breastfeeding as a binary (any/none) measure in models (presented in the main text). As most infants (98.2%) were breastfed to some extent, and it is unknown the degree to which breastfeeding, and the timing of breastfeeding, might affect infant metabolomics and lipidomics, we also considered duration of breastfeeding as a continuous variable for sensitivity analyses.

## Metabolomic and lipidomic profiling

Venous peripheral blood was collected from infants at the 6- and 12-month time points in sodium heparin and generally processed within 4 hr, with a minority (197 of 555) of 12-month samples processed after 4 hr (median time for those 197 samples = 19.9 hr, inter-quartile range [IQR] [18.7, 21.4]). The time interval between collection and post-processing storage of samples was included as a covariate in analyses. Due to the bimodal distribution of 12-month sample collection times as samples were either processed same day of collection or the following day, sensitivity analysis excluding participants with a plasma storage time greater than 4 hr (197 out of 555 infants, predominantly processed the following day) was performed, as described in the Statistical analysis section below. Plasma was stored at –80°C, and aliquots were shipped on dry ice to Nightingale Health (Helsinki, Finland) for NMR metabolomic quantification and Baker IDI (Melbourne, Australia) for liquid chromatography/ mass spectrometry (LC/MS) lipidomic quantification, as described below. For secondary analyses investigating possible 'reverse causality', that is, whether metabolomic or lipidomic profile at birth was associated with number of parent-reported infections from birth to 6 months of age, metabolomics and lipidomics data using the same platforms from venous cord blood collected at birth, as previously described (*Burugupalli et al., 2022*; *Mansell et al., 2021*), was used.

The NMR-based metabolomics platform has been described in detail (*Kettunen et al., 2016*; *Soininen et al., 2015*), and quantified a broad range of metabolic measures including lipoprotein size subclasses, triglycerides, cholesterols, fatty acids, amino acids, ketone bodies, glycolysis metabolites, and GlycA. In brief, plasma samples were mixed with a sodium phosphate buffer prior to NMR measurements with Bruker AVANCE III 500 MHz and Bruker AVANCE III HD 600 MHz spectrometers (Bruker, Billerica, MA). Samples were kept at 6°C using the SampleJet sample changer (Bruker) to prevent degradation. After initial measurements, samples went through a multiple-step lipid extraction procedure using saturated sodium chloride solution, methanol, dichloromethane, and deuterochloroform. The lipid extracts were then analysed using the 600 MHz instrument (*Soininen et al., 2015*). The utility of this platform in epidemiological research has been detailed elsewhere (*Würtz et al., 2017*). Using the Nightingale Health 2016 bioinformatics protocol, 228 metabolomic measures were generated for the 12-month samples. From a subset of participants, replicate samples were quantified, and these showed a low percentage coefficient of variation (<10%). Subsequently, the Nightingale Health 2020 bioinformatics protocol was used to generate 250 metabolomic measures for the 6-month samples, with 224 of these measures also present in the 12-month data. As a large proportion of the NMR metabolomic measures are ratios and are strongly correlated with each other in children and adults (*Ellul et al., 2019*), an informative subset of 51 measures that captured the majority of variation in the metabolomic dataset, primarily absolute metabolite concentrations, were included in analysis presented in the main text. Analyses for excluded metabolomic measures are presented as supplementary data. To complement GlycA as a measure of inflammation, hsCRP was also quantified in 6- and 12-month plasma using enzyme-linked immunosorbent assay (ELISA) (R&D Systems, Minneapolis, MN, cat. no. DY1707), as per the manufacturer's instruction.

The details of the high-performance LC/MS lipidomics platform have been described elsewhere (*Beyene et al., 2020*). In addition, we used medronic acid to passivate the LC/MS system to avoid peak tailing for acidic phospholipids (*Hsiao et al., 2018*). In brief, this platform quantified 776 lipid features in 36 lipid classes, including sphingolipids, glycerophospholipids, sterols, glycerolipids, and fatty acyls. Analysis was performed on an Agilent 6490 QQQ mass spectrometer with an Agilent 1290 series high-performance liquid chromatography system and two ZORBAX eclipse plus C18 column (2.1 × 100 mm 1.8 mm) (Agilent, Santa Clara, CA) with the thermostat set at 45°C. Mass spectrometry analysis was performed in both positive and negative ion mode with dynamic scheduled multiple reaction monitoring.

Quantification of lipid species was determined by comparison to the relevant internal standard. Lipid class total concentrations were calculated as the sum of individual lipid species concentrations, except in the case of triacylglycerols (TGs) and alkyl-diacylglycerols, where we measured both neutral loss and single ion monitoring (SIM) peaks, and subsequently used the SIM species concentrations for summation purposes.

## Statistical analysis

Analyses were performed in R (version 3.6.3) (*R Development Core Team, 2018*). All metabolomic and lipidomic measures had their lowest observed non-zero value (considered the lower limit of detection) added to their value before they were natural log-transformed and scaled to a standard distribution (standard deviation units). Pearson's correlations were calculated for number of infections from birth to 12 months with 12-month GlycA and hsCRP.

The estimated effect of number of parent-reported infections from birth to 12 months as an exposure on 12-month metabolomic and lipidomic profile was investigated using linear regression models for each metabolomic/lipidomic measure. Standard errors were used to calculate 95% confidence intervals for estimated effects. All models were adjusted for infant sex, exact age at 12-month time point, birth weight z-score, gestational age, maternal household income, exposure to maternal smoking during pregnancy, breastfeeding, and time from collection to storage for the plasma sample. Linear regression models adjusted for the same covariates were used to investigate 12-month GlycA and hsCRP as exposures and each metabolomic/lipidomic measure as an outcome. Two-tailed p-values were adjusted for multiple comparisons within each dataset (NMR metabolomics, LC/MS lipidomic species, and LC/MS lipidomic classes) using the Benjamini-Hochberg method (*Benjamini and Hochberg, 1995*). To investigate the robustness of the estimates, mean model coefficients and bias-corrected accelerated percentile bootstrap confidence intervals were calculated from nonparametric bootstrap resampling (1000 iterations) using the 'boot' package (*Davison and Hinkley, 1997*) (version 1.3–25) in R, included in Source Data. The assumption of linearity was investigated post hoc using plots of residual values for the top 10 metabolomic and lipidomic differences (ranked by p-value) for each of number of infections, GlycA, and hsCRP.

To further investigate the other sources of potential confounding and variation that could affect findings from the primary models, several sensitivity analyses were performed. These were: (i) additional adjustment of the primary model for postnatal smoking exposure, gestational diabetes, and pre-eclampsia; (ii) analyses excluding twins (five infants); (iii) analyses excluding infants with hsCRP >5 mg/L (24 infants) as a marker of acute infection (*Lemiengre et al., 2018*; *Verbakel et al., 2016*); (iv) analyses excluding plasma samples with >4 hr from collection to storage (197 samples); and (v) analyses adjusting for breastfeeding duration instead of any breastfeeding. Analyses of the primary models adjusted for different measures of socioeconomic position (SEIFA or maternal education) instead of household income were also considered. These models are presented as supplementary forest plots (*Supplementary file 1A-1F*).

Secondary analyses to investigate 6-month periods of infections were performed. The birth to 6-month models investigated the relationship between infections up to 6 months of age, 6-month inflammation, and 6-month metabolomic/lipidomic measures. The 6- to 12-month models investigated the relationship of infections between 6 and 12 months of age, 12-month inflammation, and 12-month metabolomic/lipidomic measures, with adjustment for the corresponding 6-month measures. For secondary analyses to investigate reverse causality, quasi-Poisson regression models (*Zeileis et al., 2008*) were used, either for (i) each metabolomic/lipidomic measure from cord blood at birth as exposure and total number of infections from birth to 6 months of age as outcome or (ii) each 6-month metabolomic/lipidomic measures as exposure and total number of infections from 6 to 12 months of age as the outcome, with adjustment for number of infections from birth to 6 months of age. All secondary models were adjusted for the same covariates as the primary models described above: that is, infant sex, gestational age, exact age (at 6- or 12-month time point), birth weight z-score, maternal household income, exposure to maternal smoking during pregnancy, and sample time between collection and post-processing storage. Models with birth metabolomic/lipidomic measures as the exposure were additionally adjusted for mode of delivery, which is associated with differences across many NMR metabolomic measures in cord serum in this cohort (*Mansell et al., 2021*).

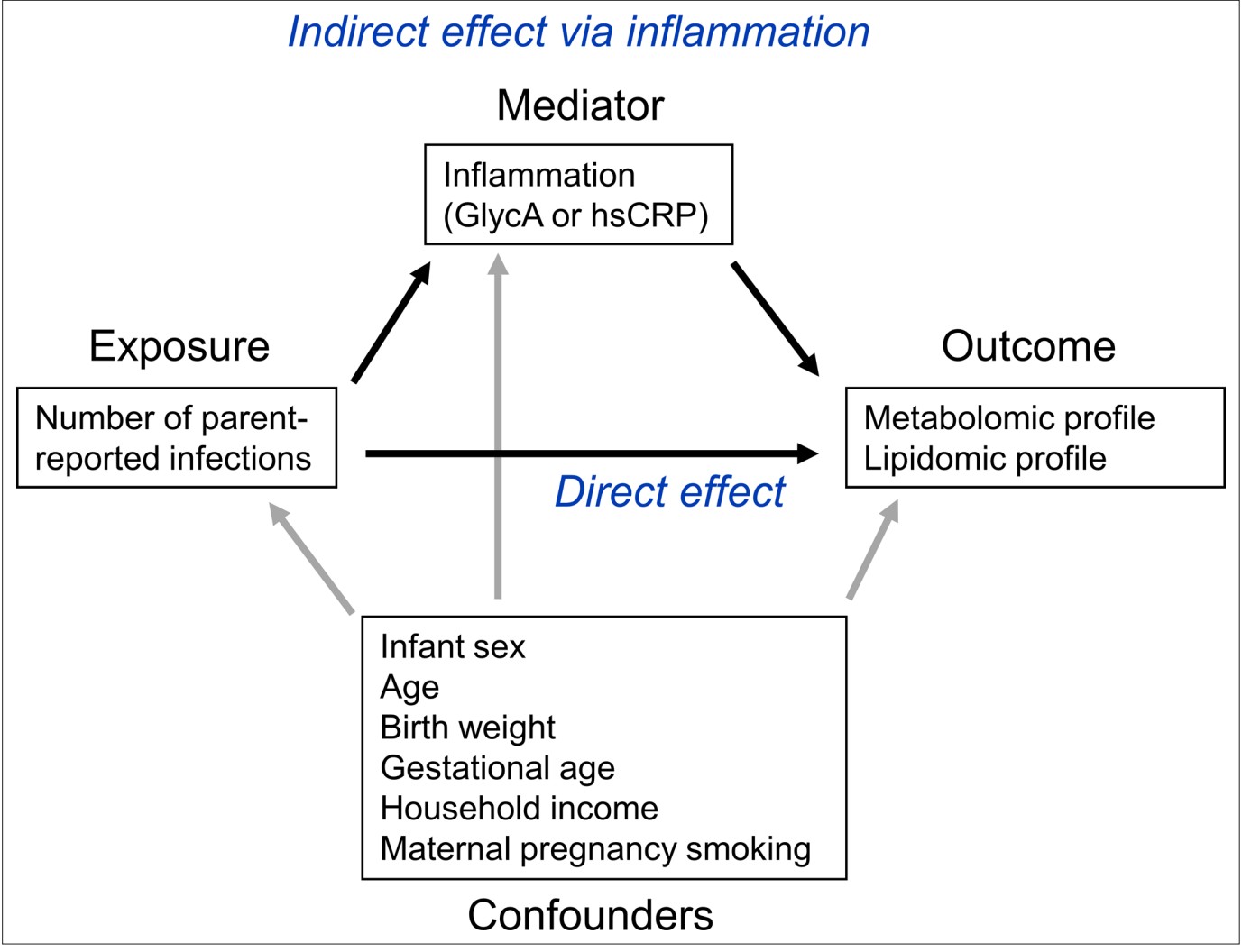

**Figure 1.** Representative directed acyclic graph (DAG) for causal model investigated in this study. The natural indirect effect (mediated by glycoprotein acetyls [GlycA] or high-sensitivity C-reactive protein [hsCRP]) and natural direct effect (not mediated by GlycA/hsCRP) of parent-reported infections on metabolomic and lipidomic measures were calculated, with adjustment for confounders. Confounders were considered to be confounders for all associations (exposure to outcome, exposure to mediator, and mediator to outcome).

To investigate the potential role of inflammation in mediating associations between infection and metabolic differences, the 'medflex' package (*Steen et al., 2017*) (version 0.6–7) in R was then used in a counterfactual-based framework to estimate the natural direct effect (not mediated by inflammation) and natural indirect effect (mediated by inflammation) of infection. Specifically, 12-month GlycA or hsCRP were considered separately as mediators for the effect of number of infections from birth to 12 months of age on metabolomic and lipidomic differences at 12 months, for all metabolomic and lipidomic measures associated with number of infections with an adjusted p-value < 0.1 from the linear regression models described above. Percentage mediation was calculated as the estimated natural indirect effect divided by the total effect (natural direct effect plus natural indirect effect). A representative directed acyclic graph (DAG) of the mediation model is shown in *Figure 1*.

As models investigating reverse causality, described above, suggested that 6-month GlycA or lipidomic measures may influence number of infections from 6 to 12 months of age, we additionally performed mediation analyses to estimate the percentage mediation of 12-month GlycA and hsCRP for the effect of number of infections from 6 to 12 months of age on 12-month metabolomic and lipidomic differences, with additional adjustment for the corresponding 6-month measures (number of infections from birth to 6 months of age, 6-month GlycA or hsCRP, and the 6-month metabolomic/lipidomic measure).

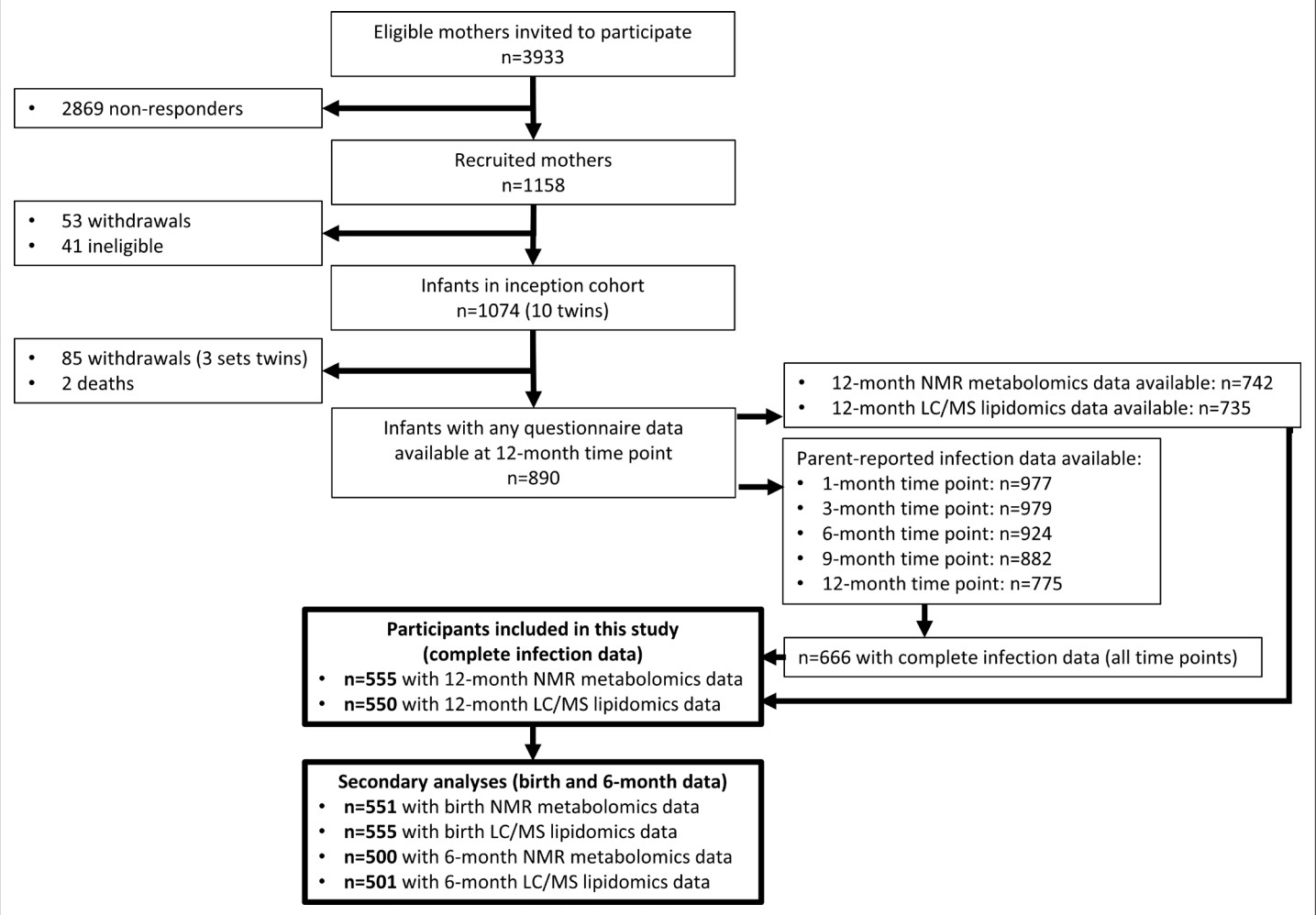

**Figure 2.** Flowchart of Barwon Infant Study participants included this study (bolded boxes). Included participants had complete infection data from all five time points between birth and 12 months of age, and 12-month plasma nuclear magnetic resonance (NMR) metabolomics data. Almost all included participants (n = 550 out of 555) had 12-month plasma liquid chromatography/mass spectrometry (LC/MS) lipidomics data.

## Results

The flowchart for the 555 infants included in this study is shown in *Figure 2*, and the cohort characteristics for these infants are shown in *Table 1*. The median number of total parent-reported infections from birth to 12 months of age was 5 (IQR = [3–8]). Median infections from birth to 6 months of age was 2 [1–3], and median infections from 6 to 12 months of age was 3 [2–5]. Median 12-month hsCRP and GlycA were 0.25 mg/L [0.08–0.96] and 1.30 mmol/L [1.16–1.48], respectively. Total number of parent-reported infections between birth and 12 months of age was more strongly correlated with 12-month GlycA (r = 0.20) than hsCRP (r = 0.11). The distributions of metabolomic and lipidomic measures at each time point for the cohort are shown in *Supplementary file 2A and B*.

### Infection and inflammation burden and plasma NMR metabolomic profile at 12 months

There was evidence for higher number of infections associating with higher inflammatory markers (GlycA (0.06 SD per 1 infection, 95% CI [0.04–0.08]) and hsCRP (0.06 [0.03–0.08])), lower HDL (−0.04 [−0.07 to −0.02]), HDL2 (−0.04 [−0.07 to −0.02]), and HDL3 (−0.04 [−0.06 to −0.01]) cholesterols, lower ApoA1 (−0.04 [−0.06 to −0.01]), lower citrate (−0.04 [−0.07 to −0.01]), higher phenylalanine (0.04 [0.02–0.07]), and to a lesser extent with higher triglycerides (0.03 [0.00–0.05]) and lower sphingomyelins (−0.03 [−0.05 to −0.01]) (*Figure 3a*). In models with GlycA as the marker of inflammation

**Table 1.** Cohort characteristics (n = 555).

| Measure | |
|---|---|
| | *n (%)* |
| Sex (female) | 269 (48.4) |
| | *Mean (SD)* |
| Maternal age at delivery (years) | 31.7 (4.5) |
| | *n (%)* |
| Maternal smoking during pregnancy (any) | 69 (12.5) |
| Gestational diabetes (cases) (n = 84 missing data) | 26 (5.5) |
| Pre-eclampsia (cases) (n = 1 missing data) | 21 (3.8) |
| Maternal annual household income (AUD) | |
| <$25,000 | 11 (2.0) |
| $25,000 to $49,999 | 41 (7.5) |
| $50,000 to $74,999 | 94 (17.2) |
| $75,000 to $99,999 | 145 (26.5) |
| $100,000 to $149,999 | 191 (34.9) |
| ≥$150,000 | 65 (11.9) |
| Maternal education (highest level completed) (n = 8 missing data) | |
| Less than year 10 of high school | 2 (0.4) |
| Year 10 of high school or equivalent | 20 (3.7) |
| Year 12 of high school or equivalent | 86 (15.7) |
| Trade/certificate/diploma | 135 (24.9) |
| Bachelor's degree | 199 (36.4) |
| Postgraduate degree | 105 (19.2) |
| Mode of birth | |
| Vaginal | 374 (67.4) |
| Planned caesarean section | 103 (18.6) |
| Unplanned caesarean section | 78 (14.1) |
| Breastfed (any breastfeeding) | 545 (98.2) |
| Infant postnatal smoke exposure to 12 months (any) (n = 47 missing data) | 14 (2.8) |
| | *Median [IQR]* |
| SEIFA index of disadvantage* | 1031 [996–1066] |
| Breastfeeding duration to 52 weeks (weeks) | 40 [16–52] |
| | *Mean (SD)* |
| Gestational age (weeks) | 39.5 (1.5) |
| Birth weight (g) | 3,538 (521) |
| Birth weight z-score | 0.32 (0.95) |
| Age at 6-month time point (months) | 6.5 (0.4) |
| Weight at 6 months (kg) | 7.9 (1.0) |
| Weight z-score at 6 months | 0.1 (1.0) |
| Age at 12-month time point (months) | 13.0 (0.8) |

*Table 1 continued on next page*

*Table 1 continued*

| Measure | |
| --- | --- |
| Weight at 12 months (kg) | 10.1 (1.3) |
| Weight z-score at 12 months | 0.4 (1.0) |

| | Median [IQR] |
| --- | --- |
| Number of parent-reported infections from birth to 12 months | 5 [3–8] |
| Infections from birth to 6 months | 2 [1–3] |
| Infection from 6 to 12 months | 3 [2–5] |

| | |
| --- | --- |
| GlycA at 6 months (mmol/L) | 0.76 [0.68–0.85] |
| hsCRP at 6 months (mg/L) | 0.14 [0.05–0.94] |
| GlycA at 12 months (mmol/L) | 1.30 [1.16–1.48] |
| hsCRP at 12 months (mg/L) | 0.25 [0.08–0.96] |

All n = 555 infants had complete covariate data for primary models, missing data of secondary exposures is indicated next to the relevant measure.
*A lower SEIFA value indicates greater socioeconomic disadvantage based on postcode.

burden, metabolomic differences observed for higher GlycA were largely similar to, but more marked than, those for parent-reported infections; including cholesterols (lower HDL (−0.38 SD per 1 SD increase in log GlycA [−0.46 to −0.30]), higher LDL (0.18 [0.09–0.26]), and very-large-density lipoprotein (0.50 [0.42–0.58]) cholesterols), apolipoproteins (lower ApoA1 (−0.23 [−0.31 to −0.14]), higher apolipoprotein B (ApoB) (0.39 [0.31–0.48])), higher total fatty acids (0.36 [0.28–0.45]), higher total triglycerides (0.48 [0.40–0.56]) and cholines (0.18 [0.10–0.27]), amino acids (higher phenylalanine (0.17 [0.09–0.25]), isoleucine (0.14 [0.05–0.22]), and glycine (0.16 [0.08–0.23]), lower histidine (−0.16 [−0.24 to −0.09])), glycolysis-related metabolites (higher pyruvate (0.06 [0.01–0.11]), lower citrate (−0.17 [−0.26 to −0.08]), and lower acetoacetate (−0.18 [−0.26 to −0.09]) (*Figure 3b*)). Bootstrap estimates were generally similar to standard regression estimates for all models. Estimated effect sizes were generally similar across most sensitivity analyses (*Supplementary file 1A-C*), though excluding samples with processing time greater than 4 hr slightly reduced the magnitude of estimated effects of parent-reported infections on HDL cholesterols and ApoA1 (*Supplementary file 1A*). Higher hsCRP was associated with lower HDL cholesterol (−0.24 SD per 1 SD increase in log hsCRP, [−0.32 to −0.16]), ApoA1 (−0.26 [−0.34 to −0.18]) and histidine (−0.20 [−0.27 to −0.12]), and higher phenylalanine (0.14 [0.06 to 0.22]), as observed for GlycA, and with lower levels of most other amino acids and albumin (−0.11 [−0.19 to −0.03]), and higher LDL triglycerides (0.16 [0.08–0.25]) (*Figure 4a*).

There was a stronger correlation for the metabolomic differences related to infection and those related to GlycA (r = 0.74, *Figure 3c*) than for infection and hsCRP (r = 0.62, *Figure 4b*). The correlation of metabolomic differences related to GlycA and those for hsCRP was r = 0.61 (*Figure 4c*).

In models investigating the relationship between number of parent-reported infections from birth to 6 months of age and 6-month metabolomic profile, there was evidence of a similar, but less marked, pattern of associations to that seen at 12 months of age. Higher number of infections was modestly associated with higher inflammatory markers (GlycA (0.07 SD per 1 infection, [0.02–0.12]) and hsCRP (0.06 [0.01–0.11])), lower HDL cholesterol (−0.06 [−0.11 to −0.01]), and lower ApoA1 (−0.07 [−0.12 to −0.02]) (*Figure 3—figure supplement 1a*). At 6 months of age, higher GlycA was associated with a similar pattern of 6-month metabolomic differences as seen at 12 months (*Figure 3—figure supplement 1b*), as was hsCRP (*Figure 4—figure supplement 1a*). In models investigating the relationship between number of parent-reported infections from 6 to 12 months of age and 12-month metabolomic profile (i.e. infections within the preceding 6 months) with adjustment for birth to 6-month infections, associations between number of infections and metabolomic profile were similar to models considering number of infections from birth to 12 months of age (*Figure 3—figure supplement 2a*). Models considering 12-month GlycA or hsCRP as the exposure with adjustment for 6-month GlycA or

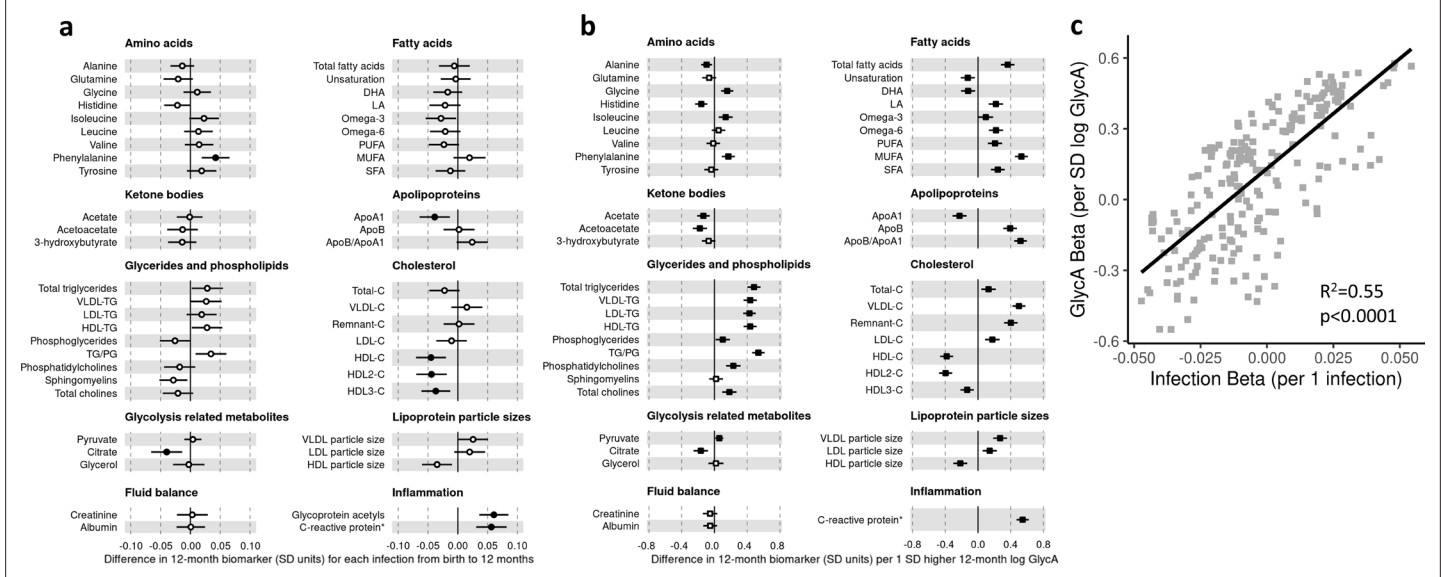

**Figure 3.** Difference in 12-month plasma nuclear magnetic resonance (NMR) metabolomic measures for each increase in parent-reported infection (birth to 12 months) and for each SD increase in 12-month glycoprotein acetyls (GlycA) (n = 555). Forest plots of the estimated 12-month metabolomic differences for each additional parent-reported infection from birth to 12 months (**a**, circle points) or SD log 12-month GlycA (**b**, square points) from adjusted linear regression models, and the correlation of estimated metabolomic differences for these two exposures (**c**). Error bars are 95% confidence intervals. Closed points represent adjusted p-value < 0.05. All models were adjusted for infant age, sex, gestational age, birth weight, maternal household income, smoking during pregnancy, breastfeeding status, and sample processing time. Infection and GlycA exposure model estimates and details for all NMR metabolomic measures are shown in *Figure 3—source data 1*.

The online version of this article includes the following source data and figure supplement(s) for figure 3:

**Source data 1.** Summary of regression models for difference in 12-month nuclear magnetic resonance (NMR) metabolomic measures per 1 increase in parent-reported infection from birth to 12 months of age or per SD increase in 12-month log glycoprotein acetyls (GlycA).

**Source data 2.** Summary of regression models for difference in 6-month nuclear magnetic resonance (NMR) metabolomic measures per 1 increase in parent-reported infection from birth to 6 months of age or per SD increase in 6-month log GlycA.

**Source data 3.** Summary of regression models for difference in 12-month nuclear magnetic resonance (NMR) metabolomic measures (adjusted for corresponding 6-month measure) per 1 increase in parent-reported infection from 6 to 12 months of age (adjusted for infections from birth to 6 months) or per SD increase in 12-month log glycoprotein acetyls (GlycA) (adjusted for 6-month GlycA).

**Figure supplement 1.** Difference in 6-month plasma nuclear magnetic resonance (NMR) metabolomic measures for each increase in parent-reported infection from birth to 6 months and for each SD increase in 6-month log glycoprotein acetyls (GlycA) (n = 500).

**Figure supplement 2.** Difference in 12-month plasma nuclear magnetic resonance (NMR) metabolomic measures (adjusted for corresponding 6-month measure) per 1 increase in parent-reported infection from 6 to 12 months of age (adjusted for infections from birth to 6 months) or per SD increase in 12-month log glycoprotein acetyls (GlycA) (adjusted for 6-month GlycA) (n = 500).

hsCRP, respectively, also resembled those without adjustment for 6-month inflammation (*Figure 3— figure supplement 2b*, *Figure 4—figure supplement 2a*).

In secondary analyses, there was little evidence for associations between serum NMR metabolomic measures at birth at number of parent-reported infections from birth to 6 months of age (*Supplementary file 3A*). Similarly, there was little evidence for metabolomic measures at 6 months of age associating with number of infections from 6 to 12 months of age. GlycA at 6 months showed a modest association with a higher number of infections from 6 to 12 months (average 0.10 higher infections per 1 SD higher log 6-month GlycA, [0.04–0.16]), adjusted for the number of infections from birth to 6 months; for hsCRP there was little evidence (0.02 [−0.04–0.08]) (*Supplementary file 3B*).

## Infection and inflammation burden and plasma LC/MS lipidomic profile at 12 months

In regression models with number of parent-reported infections as exposure and LC/MS lipids as the outcomes, infants with more infections had, on average, lower levels of the dehydrocholesteryl ester (−0.05 SD per 1 infection, [−0.08 to −0.03]) and trihexosylceramide (−0.04 [−0.06 to −0.01]) class lipids.

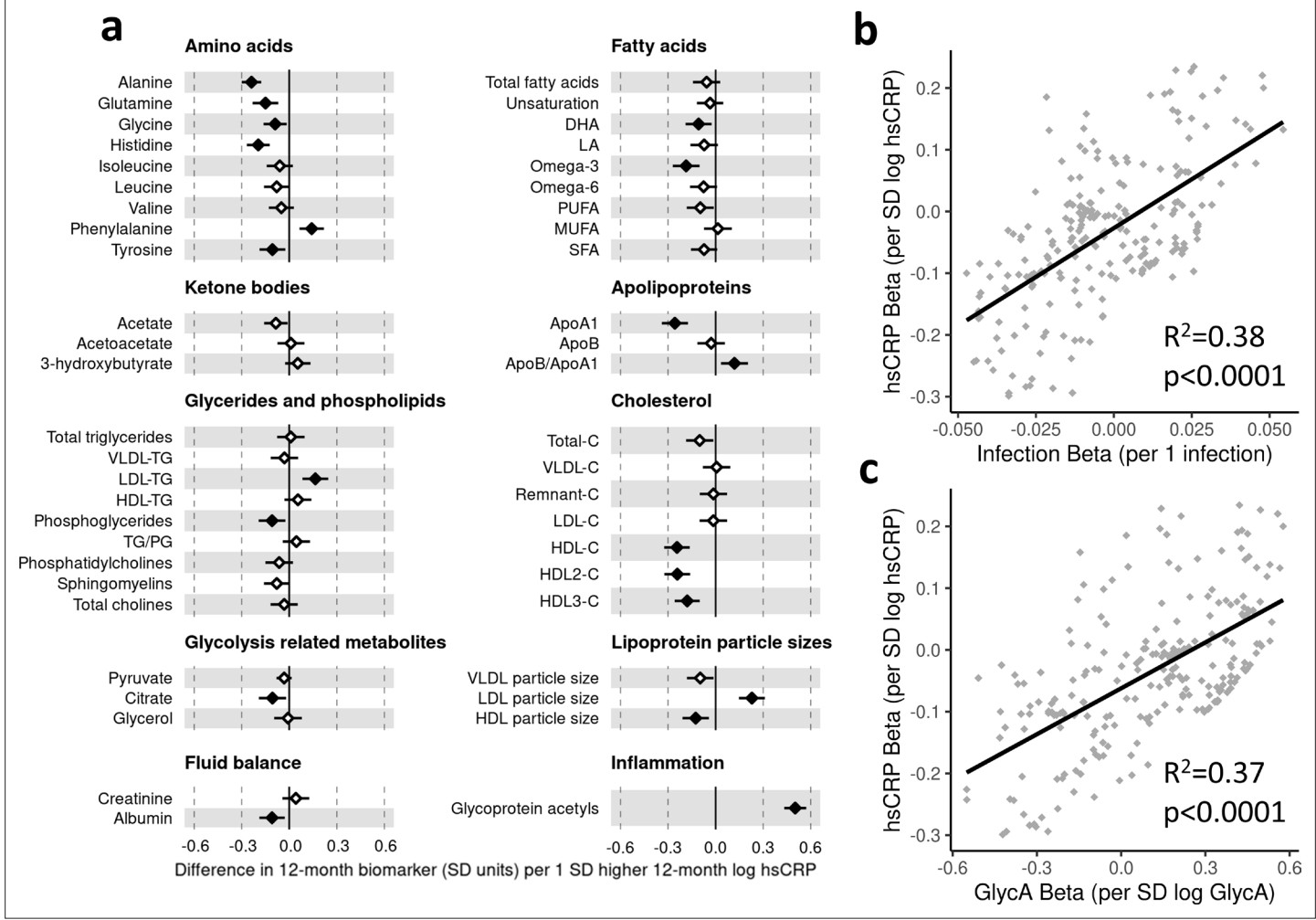

**Figure 4.** Difference in 12-month plasma nuclear magnetic resonance (NMR) metabolomic measures for each SD increase in 12-month high-sensitivity C-reactive protein (hsCRP) (n = 555). Forest plot for the estimated 12-month metabolomic differences for each additional SD log 12-month hsCRP (**a**, diamond points) from adjusted linear regression models, and the correlation of estimated metabolomic differences for infection and hsCRP (**b**) and for glycoprotein acetyls (GlycA) and hsCRP (**c**). Error bars are 95% confidence intervals. Closed points represent adjusted p-value < 0.05. All models were adjusted for infant age, sex, gestational age, birth weight, maternal household income, smoking during pregnancy, breastfeeding status, and sample processing time. hsCRP exposure model estimates and details for all NMR metabolomic measures are shown in *Figure 4—source data 1*.

The online version of this article includes the following source data and figure supplement(s) for figure 4:

**Source data 1.** Summary of regression models for difference in 12-month nuclear magnetic resonance (NMR) metabolomic measures per SD increase in 12-month log high-sensitivity C-reactive protein (hsCRP).

**Source data 2.** Summary of regression models for difference in 6-month nuclear magnetic resonance (NMR) metabolomic measures per SD increase in 6-month log high-sensitivity C-reactive protein (hsCRP).

**Source data 3.** Summary of regression models for difference in 12-month nuclear magnetic resonance (NMR) metabolomic measures (adjusted for corresponding 6-month measure) per SD increase in 12-month log high-sensitivity C-reactive protein (hsCRP) (adjusted for 6-month hsCRP).

**Figure supplement 1.** Difference in 6-month plasma nuclear magnetic resonance (NMR) metabolomic measures for each SD increase in 6-month log high-sensitivity C-reactive protein (hsCRP) (n = 500).

**Figure supplement 2.** Difference in 12-month plasma nuclear magnetic resonance (NMR) metabolomic measures (adjusted for corresponding 6-month measure) per SD increase in 12-month log high-sensitivity C-reactive protein (hsCRP) (adjusted for 6-month hsCRP) (n = 500).

There was also evidence to a lesser extent for associations between higher number of infections and lower cholesteryl esters (−0.03 [−0.05 to −0.01]) and plasmalogen classes (lysoalkenyl phosphatidylcholines, −0.02 [−0.04–0.00]; alkenylphosphatidylethanolamines, −0.03 [−0.06 to −0.01]) (*Figure 5a*). The 10 lipid species with the strongest statistical evidence for association with number of infections were hexosylceramides (HexCer(d18:2/18:0), −0.05 [−0.07 to −0.02]; HexCer(d18:2/22:0), −0.04 [−0.07 to

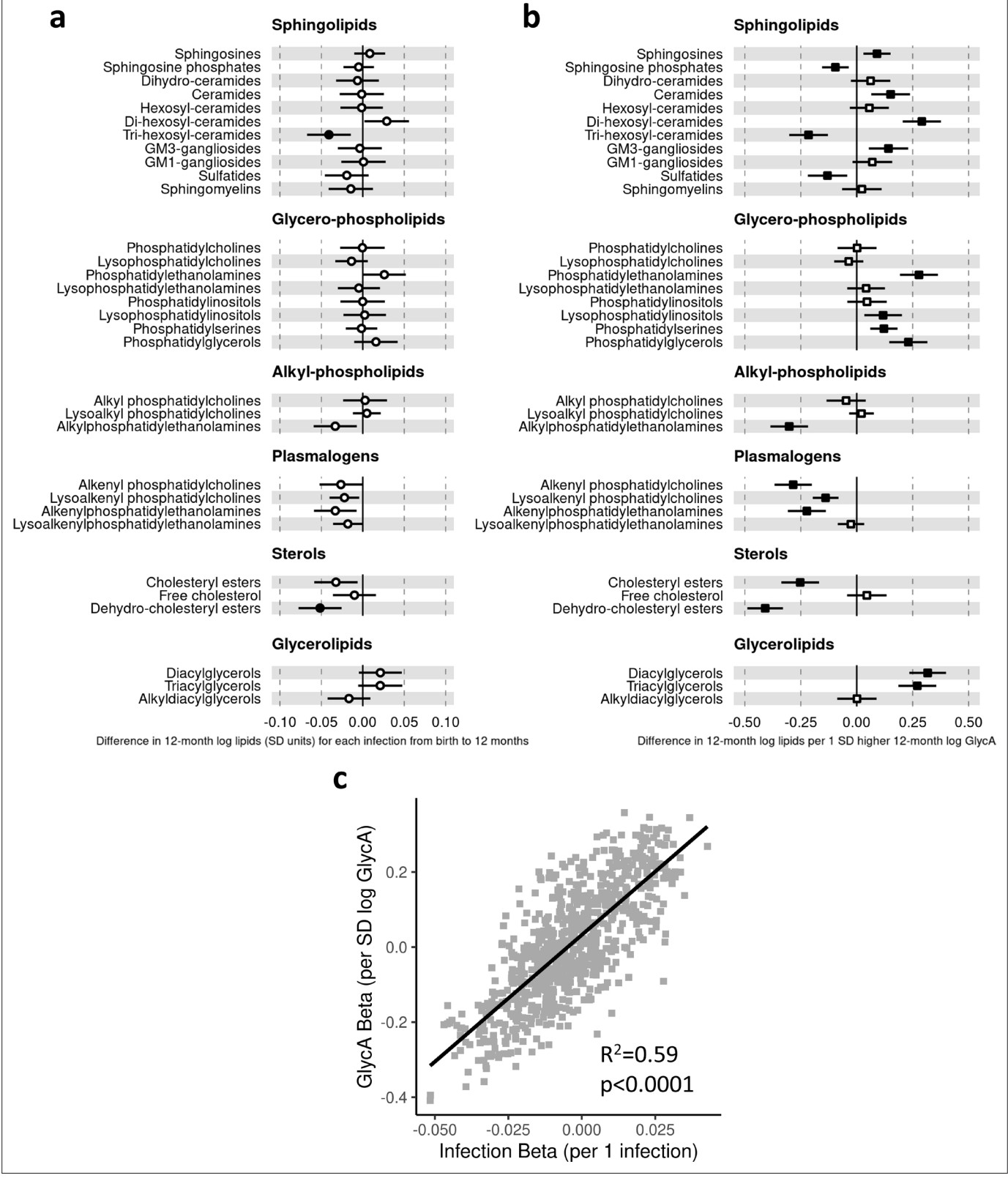

**Figure 5.** Difference in 12-month plasma liquid chromatography/mass spectrometry (LC/MS) lipidomic class totals for each increase in parent-reported infection (birth to 12 months) and for each SD increase in 12-month glycoprotein acetyls (GlycA) (n = 550). Forest plots of the estimated 12-month lipidomic differences in class totals for each additional parent-reported infection from birth to 12 months (**a**, circle points) or SD log 12-month GlycA (**b**, square points) from adjusted linear regression models, and the correlation of estimated differences for these two exposures across all lipidomic

*Figure 5 continued on next page*

*Figure 5 continued*

measures (**c**). In (**a**) and (**b**), error bars are 95% confidence intervals. Closed points represent adjusted p-value < 0.05. All models were adjusted for infant age, sex, gestational age, birth weight, maternal household income, smoking during pregnancy, breastfeeding status, and sample processing time. Forest plots depicting individual lipid species within each group are shown in *Figure 5—figure supplement 1*. Infection and GlycA exposure model estimates and details for all LC/MS lipidomic measures are shown in *Figure 5—source data 1*.

The online version of this article includes the following source data and figure supplement(s) for figure 5:

**Source data 1.** Summary of regression models for difference in 12-month liquid chromatography/mass spectrometry (LC/MS) lipidomic measures per 1 increase in parent-reported infection from birth to 12 months of age or per SD increase in 12-month log glycoprotein acetyls (GlycA).

**Source data 2.** Summary of regression models for difference in 6-month liquid chromatography/mass spectrometry (LC/MS) lipidomic measures per 1 increase in parent-reported infection from birth to 6 months of age or per SD increase in 6-month log glycoprotein acetyls (GlycA).

**Source data 3.** Summary of regression models for difference in 12-month liquid chromatography/mass spectrometry (LC/MS) lipidomic measures (adjusted for corresponding 6-month measure) per 1 increase in parent-reported infection from 6 to 12 months of age (adjusted for infections from birth to 6 months) or per SD increase in 12-month log glycoprotein acetyls (GlycA) (adjusted for 6-month GlycA).

**Figure supplement 1.** Forest plots showing the difference in 12-month liquid chromatography/mass spectrometry (LC/MS) lipidomic classes and lipid species per 1 increase in parent-reported infection from birth to 12 months of age and per SD increase in 12-month log glycoprotein acetyls (GlycA) (n = 550).

**Figure supplement 2.** Difference in 6-month plasma liquid chromatography/mass spectrometry (LC/MS) lipidomic measures for each increase in parent-reported infection from birth to 6 months and for each SD increase in 6-month log glycoprotein acetyls (GlycA) (n = 501).

**Figure supplement 3.** Difference in 12-month plasma liquid chromatography/mass spectrometry (LC/MS) lipidomic measures (adjusted for corresponding 6-month measure) per 1 increase in parent-reported infection from 6 to 12 months of age (adjusted for infections from birth to 6 months) or per SD increase in 12-month log glycoprotein acetyls (GlycA) (adjusted for 6 month GlycA) (n = 496).

−0.02]; HexCer(d16:1/24:0), −0.04 [−0.07 to −0.02]), trihexosylceramides (Hex3Cer(d18:1/18:0), −0.04 [−0.07 to −0.02]; Hex3Cer(d18:1/20:0), −0.05 [−0.07 to −0.02]; Hex3Cer(d18:1/22:0), −0.04 [−0.07 to −0.02]), phosphatidylethanolamines (PE(18:0/20:3), 0.04 [0.02–0.07]), cholesteryl esters (CE(22:5), −0.04 [−0.07 to −0.02]; CE(22:6), −0.04 [−0.07 to −0.02]) and dehydrocholesteryl esters (DE(18:2), −0.05 [−0.08 to −0.03]) (*Figure 5—figure supplement 1a*).

Compared to models with number of infections, lipidomic differences were more pronounced when GlycA was considered as the exposure, including higher ceramides (0.15 SD per 1 SD increase in log GlycA, [0.06–0.24]), dihexosylceramides (0.29 [0.21–0.38]), di- (0.32 [0.24–0.40]) and TGs (0.27 [0.19–0.36]), and phospholipid classes (e.g. phosphatidylethanolamines, 0.28 [0.19–0.36]; phosphatidylglycerols, 0.23 [0.15–0.32]), and lower plasmalogen classes (alkenyl phosphatidylcholines, −0.28 [−0.37 to −0.21]; alkenylphosphatidylethanolamines, −0.22 [−0.31 to −0.14]; lysoalkenyl phosphatidylcholines, −0.14 [−0.20 to −0.08]), cholesteryl esters (−0.25 [−0.34 to −0.17]), and dehydrocholesteryl esters (−0.41 [−0.49 to −0.33]) (*Figure 5b*).

Higher hsCRP was also associated with differences across several lipid classes, particularly lower plasmalogen classes (e.g. lysoalkenyl phosphatidylcholines, −0.19 SD per 1 SD increase in log hsCRP, [−0.25 to −0.14]; alkenyl phosphatidylcholines, −0.26 [−0.35 to −0.18]), sulfatides (−0.26 [−0.34 to −0.17]), and alkyl phosphatidylcholines (−0.29 [−0.38 to −0.21]) (*Figure 6a*). For all lipidomic models, bootstrap estimates were similar to the standard regression estimates. All sensitivity analyses (*Supplementary file 1D-1F*) showed similar estimated effect sizes, except for hsCRP models excluding samples with processing times greater than 4 hr, where estimated differences in several classes (including lysophosphatidylcholines, lysoalkenyl phosphatidylcholines, and lysoalkenylphosphatidylethanolamines) were substantially larger compared to the primary models (*Supplementary file 1F*).

There was stronger correlation for lipidomic differences related to infection and GlycA (r = 0.77, *Figure 5c*) than for infection and hsCRP (r = 0.36, *Figure 6b*) or GlycA and hsCRP (r = 0.55, *Figure 6c*).

In models investigating parent-reported infections from birth to 6 months of age and 6-month lipidomic measures, a similar, but less marked, pattern of associations was seen as at 12 months (*Figure 5—figure supplement 2a*). Models of 6-month GlycA (*Figure 5—figure supplement 2b*) or 6-month hsCRP (*Figure 6—figure supplement 2a*) and 6-month lipidomic measures were largely the same as the 12-month models with some exceptions, such as higher GlycA associating with lower GM3-gangliosides at 6 months (−0.10 SD per 1 SD higher log GlycA, 95% CI [−0.18 to −0.02]) in contrast to associating with higher GM3-gangliosides at 12 months (0.14 [0.05–0.23]). In models investigating infections from 6 to 12 months of age and 12-month lipidomic measures, with adjustment for infection from birth to 6 months, a similar pattern of associations was seen as in models for number of

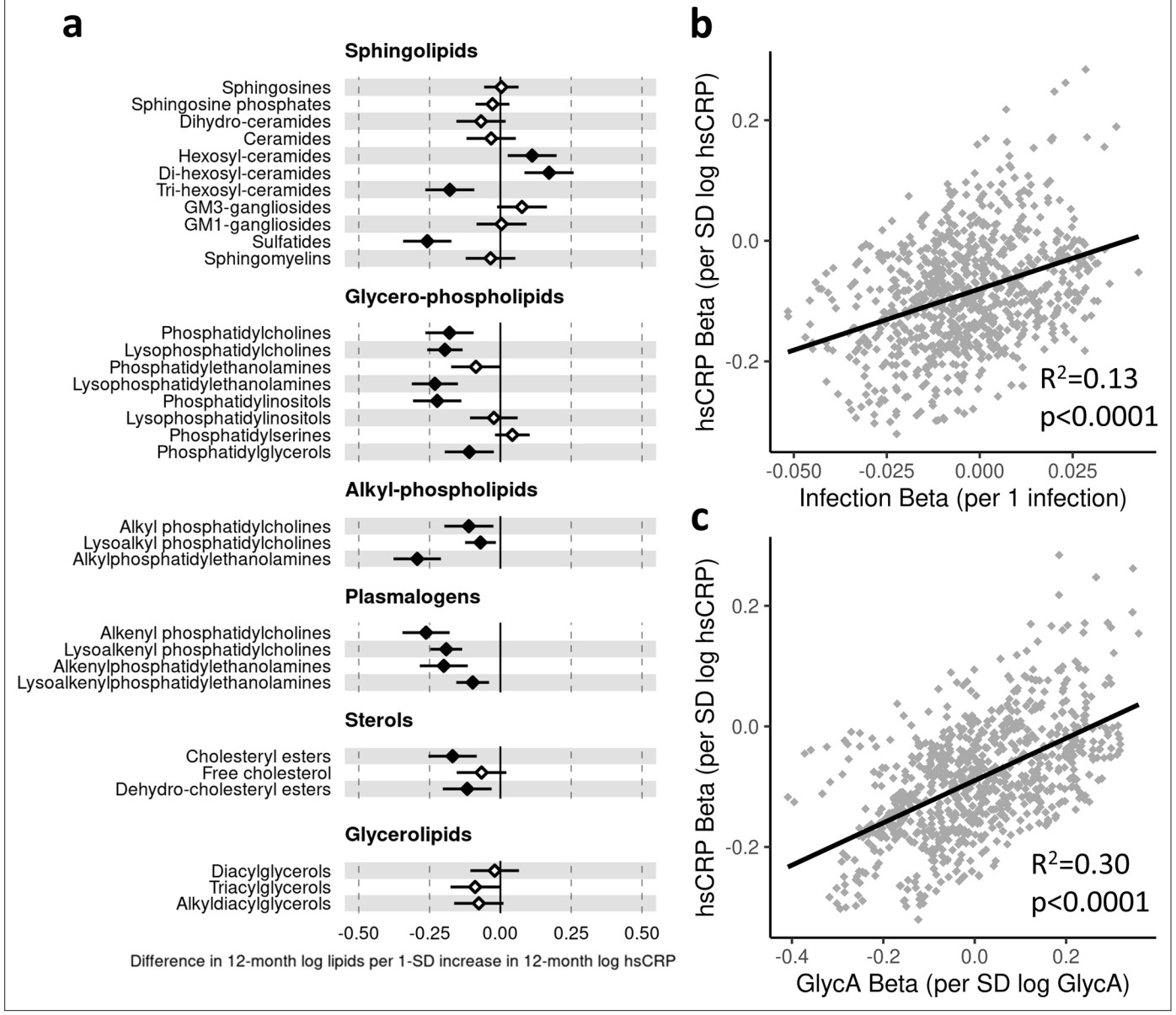

**Figure 6.** Difference in 12-month plasma liquid chromatography/mass spectrometry (LC/MS) lipidomic class totals for each SD increase in 12-month high-sensitivity C-reactive protein (hsCRP) (n = 550). Forest plot for the estimated 12-month lipidomic differences for each additional SD log 12-month hsCRP (**a**, diamond points) from adjusted linear regression models, and the correlation of estimated differences across all lipidomic measures for infection and hsCRP (**b**) and glycoprotein acetyls (GlycA) and hsCRP (**c**). In (**a**), error bars are 95% confidence intervals. Closed points represent adjusted p-value < 0.05. All models were adjusted for infant age, sex, gestational age, birth weight, maternal household income, smoking during pregnancy, breastfeeding status, and sample processing time. Forest plots depicting individual lipid species within each group are shown in *Figure 6—figure supplement 1*. hsCRP exposure model estimates and details for all LC/MS lipidomic measures are shown in *Figure 6—source data 1*.

The online version of this article includes the following source data and figure supplement(s) for figure 6:

**Source data 1.** Summary of regression models for difference in 12-month liquid chromatography/mass spectrometry (LC/MS) lipidomic measures per SD increase in 12-month log high-sensitivity C-reactive protein (hsCRP).

**Source data 2.** Summary of regression models for difference in 6-month liquid chromatography/mass spectrometry (LC/MS) lipidomic measures per SD increase in 6-month log high-sensitivity C-reactive protein (hsCRP).

**Source data 3.** Summary of regression models for difference in 12-month liquid chromatography/mass spectrometry (LC/MS) lipidomic measures (adjusted for corresponding 6-month measure) per SD increase in 12-month log high-sensitivity C-reactive protein (hsCRP) (adjusted for 6-month hsCRP).

**Figure supplement 1.** Forest plots showing the difference in 12-month liquid chromatography/mass spectrometry (LC/MS) lipidomic classes and lipid species per SD increase in 12-month log high-sensitivity C-reactive protein (hsCRP) (n = 550).

*Figure 6 continued on next page*

Figure 6 continued

**Figure supplement 2.** Difference in 6-month plasma liquid chromatography/mass spectrometry (LC/MS) lipidomic measures for each SD increase in 6-month log high-sensitivity C-reactive protein (hsCRP) (n = 501).

**Figure supplement 3.** Difference in 12-month plasma liquid chromatography/mass spectrometry (LC/MS) lipidomic measures (adjusted for corresponding 6-month measure) per SD increase in 12-month log high-sensitivity C-reactive protein (hsCRP) (adjusted for 6-month hsCRP) (n = 496).

infections from birth to 12 months (*Figure 5—figure supplement 3a*). Likewise, models for 12-month GlycA or hsCRP and 12-month lipidomic measures showed very similar results with adjustment for 6-month measures compared to models without 6-month adjustment (*Figure 5—figure supplement 3b*, *Figure 6—figure supplement 3a*).

As with the NMR metabolomic measures, there was little evidence for associations between serum LC/MS lipidomic measures at birth and number of infections from birth to 6 months of age in secondary analyses (*Supplementary file 3C*). For 6-month measures, higher cholesteryl esters and dehydrocholesteryl esters were associated with lower number of subsequent infections from 6 to 12 months of age (average −0.10 lower infections per 1 SD higher 6-month log total cholesteryl esters, 95% CI [−0.15 to −0.05]; −0.10 infections per 1 SD higher 6-month log total dehydrocholesteryl esters, 95% CI [−0.16 to −0.05]) (*Supplementary file 3D*).

## Mediation analysis

We next assessed whether inflammation (i.e. GlycA or hsCRP at 12 months) mediated the effects of infection on specific metabolite and lipid measures (adjusted p-value < 0.1) (*Figure 7*). For the NMR metabolomic measures considered in mediation models, there was evidence of an indirect effect of infection mediated by GlycA on all measures, and an indirect effect mediated by hsCRP for phenylalanine, ApoA1, and the HDL, HDL2, and HDL3 cholesterols. For all measures except ApoA1 and HDL3 cholesterol, GlycA was estimated to mediate a larger proportion of the total effect of infections on metabolomic measures than hsCRP. For the LC/MS lipidomic mediation models, there was evidence of indirect effects of infection on all the considered lipid classes mediated by GlycA, and indirect effects mediated by hsCRP for all classes except dehydrocholesteryl esters. GlycA was estimated to mediate a larger proportion of the total effect on infections on all lipid classes except lysoalkenyl phosphatidylcholines. GlycA was similarly estimated to mediate a larger proportion of the total effect of infection on individual lipid species than hsCRP for all lipid species included in mediation analyses.

To consider potential confounding from earlier inflammation or metabolomic/lipidomic measures, we additionally performed mediation analyses for inflammation measured at 12 months mediating the relationship between number of infections from 6 to 12 months of age and the 12-month metabolomic/lipidomic measures with adjustment for number of infections from birth to 6 months of age, 6-month inflammation, and the corresponding 6-month metabolomic/lipidomic measure. For most metabolomic and lipidomics measures, evidence for indirect effects mediated by GlycA or hsCRP varied minimally in this 6- to 12-month model compared to the birth to 12-month mediation models (*Figure 7—figure supplement 1*).

## Discussion

In this study, cumulative parent-reported infection burden from birth to 12 months was associated with adverse NMR metabolomic and LC/MS lipidomic profiles at 12 months of age. Similar but more marked effects on these profiles were evident when considering GlycA, a cumulative inflammation marker, as an exposure. In contrast, differences in metabolomic and lipidomic profiles associated with higher hsCRP were largely distinct from, and less marked than, those for GlycA, suggesting that GlycA may be superior to hsCRP as an early life marker of infection burden. There was evidence that inflammation (with GlycA generally showing stronger evidence as a mediator than hsCRP) may partly mediate many of the largest metabolomic and lipidomic differences.

These findings are novel and of potentially considerable significance; the burden of infection falls largely in infancy and early childhood, and these are among the first data to explore the cardiometabolic associations with common infections in this age group. Cardiometabolic risk accrues across the entire life course; the American Heart Association states that 'with primordial and primary prevention, CVD is largely preventable' and that risk stratification and intervention are more likely to be successful

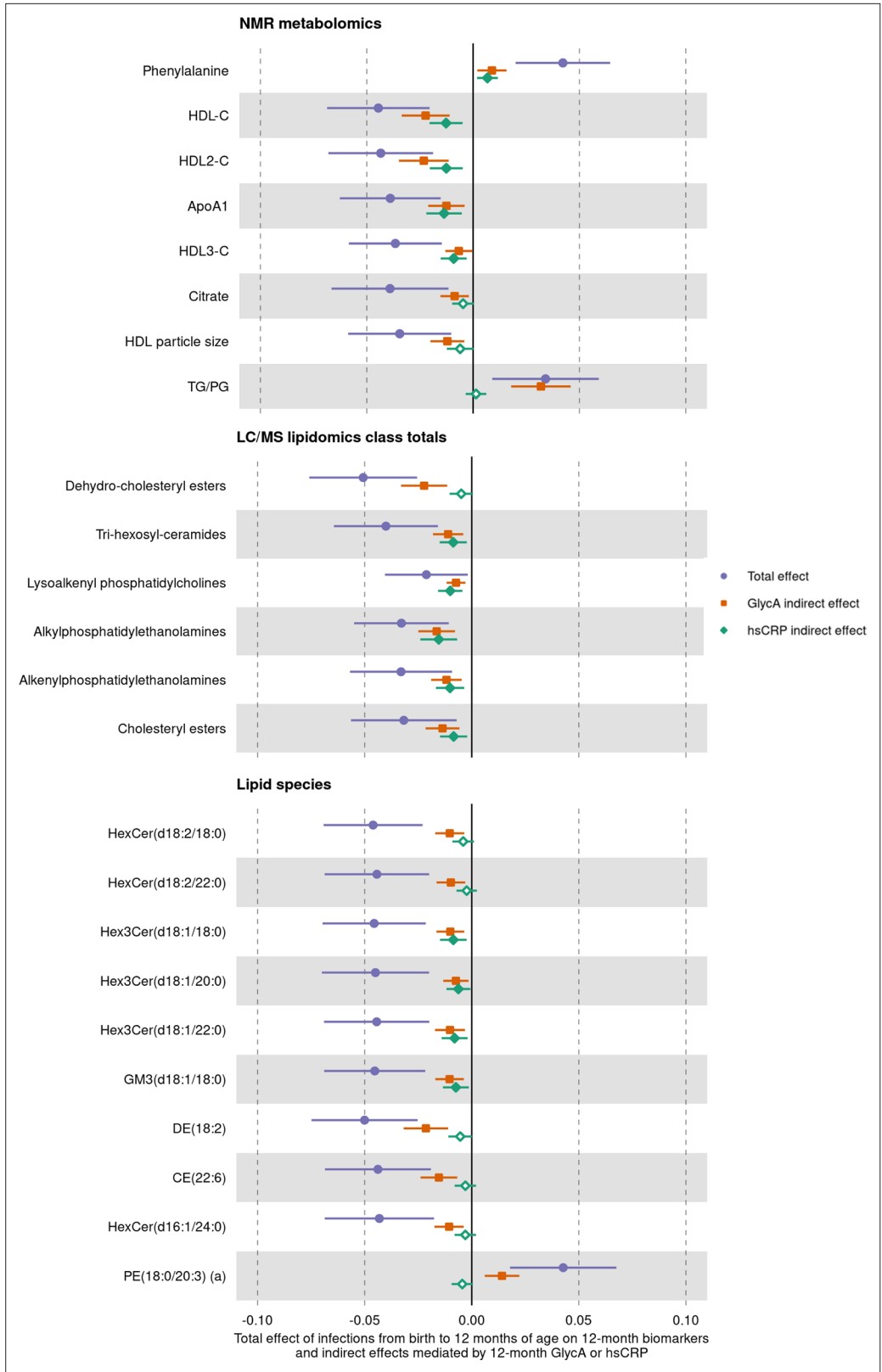

**Figure 7.** Total effect of infection on 12-month metabolomic and lipidomic measures (purple, circle points) and the estimated natural indirect effect component of these mediated by glycoprotein acetyls (GlycA) (orange, square points) or high-sensitivity C-reactive protein (hsCRP) (green, diamond points). Units of change are 1 infection for parent-reported infections, and 1 SD change for GlycA, hsCRP, and metabolomic/lipidomic measures on log scale.

*Figure 7 continued on next page*

*Figure 7 continued*

Error bars are 95% confidence intervals. Closed points represent p-value < 0.05. All models were adjusted for infant age, sex, gestational age, birth weight, maternal household income, smoking during pregnancy, breastfeeding status, and sample processing time. Model details are in *Figure 7—source data 1*.

The online version of this article includes the following source data and figure supplement(s) for figure 7:

**Source data 1.** Summary of mediation models for total, natural direct, and natural indirect effects (mediated by glycoprotein acetyls [GlycA] or high-sensitivity C-reactive protein [hsCRP]) of infection from birth to 12 months on 12-month metabolomic and lipidomic measures.

**Source data 2.** Summary of mediation models for total, natural direct, and natural indirect effects (mediated by glycoprotein acetyls [GlycA] or high-sensitivity C-reactive protein [hsCRP]) of infection from 6 to 12 months on 12-month metabolomic and lipidomic measures, with adjustment for infections from birth to 6 months of age, 6-month inflammation, and the corresponding 6-month metabolomic/lipidomic measure.

**Figure supplement 1.** Total effect of infection from 6 to 12 months on 12-month metabolomic and lipidomic measures (purple, circle points) and the estimated natural indirect effect component of these mediated by 12-month glycoprotein acetyls (GlycA) (orange, square points) or high-sensitivity C-reactive protein (hsCRP) (green, diamond points), with adjustment for infections from birth to 6 months of age, 6-month inflammation, and the corresponding 6-month metabolomic/lipidomic measure.

if done earlier in life (*Weintraub et al., 2011*). Therefore, understanding associations in early life in general populations is an important step towards risk stratification and targeted interventions. There are few data investigating LC/MS lipidomics in response to common infections in high-income countries; indeed some of the only analogous studies investigated lipidomics following Ebola (*Kyle et al., 2019*) or acute lower respiratory tract infection (*Gao et al., 2019*) in adults. These are the only analyses to explore the potential mediation of these associations between infections and omics profiles by inflammation.

The metabolomic profiles observed in this study in infants with greater infection burden (higher triglycerides, lower HDL cholesterol, and lower ApoA1) reflect those previously linked to infection, including SARS-CoV-2 (*Alvarez and Ramos, 1986*; *Bruzzone et al., 2020*; *Gallin et al., 1969*; *Liuba et al., 2003*; *Madsen et al., 2018*), and CVD risk (particularly higher phenylalanine) in adults (*Würtz et al., 2015*; *Würtz et al., 2012*). Acute and chronic infection-related metabolic differences have been implicated in atherosclerosis (*Khovidhunkit et al., 2000*), including increased carotid intima-media thickness (*Burgner et al., 2015c*; *Liuba et al., 2003*) and arterial stiffness (*Charakida et al., 2009*) in older children, and in adult CVD risk (*Bergh et al., 2017*; *Burgner et al., 2015a*). The mechanisms underpinning the link between inflammation and metabolism are not well characterised, but there is evidence they may share regulatory pathways. For example, lipid-activated nuclear receptors such as peroxisome-proliferator-activated receptors and liver X receptors regulate both lipid metabolism and inflammation (*Bensinger and Tontonoz, 2008*). Synthesis of leptin, a key regulatory hormone of metabolism, including energy homeostasis (*Park and Ahima, 2015*), is increased by proinflammatory cytokines during acute infection (*Behnes et al., 2012*), and is in turn implicated in many inflammatory processes (*Abella et al., 2017*).

In lipidomic analyses, infection burden was associated with lower levels of several ether phosphatidylethanolamine species (PC(O) and PC(P)), consistent with the reported decrease in these polyunsaturated-fatty acid (PUFA)-containing species in SARS-CoV-2 infected adults (*Schwarz et al., 2020*). Importantly, these decreases contrast with observed increases in diacyl-phosphatidylethanolamine species, and this discordance in phosphatidylethanolamine species is characteristic of conditions characterised by inflammation, including Alzheimer's disease (*Huynh et al., 2020*). The role of phosphatidylethanolamines in infection is incompletely understood, but omega-3 fatty acids (a major class of PUFA) have been implicated in anti-inflammatory pathways (*Calder, 2013*) and ether phosphatidylethanolamine species are a potential source of PUFAs. Lower trihexosylceramide lipid class, associated with more infections, has been linked to increased BMI and pre-diabetic phenotypes in adults (*Meikle et al., 2013*; *Weir et al., 2013*).

The different relationships of the two inflammatory markers GlycA and hsCRP with metabolomic and lipidomic profiles are consistent with a recent study in pregnant women reporting GlycA was more strongly correlated with NMR metabolomic differences than hsCRP (*Mokkala et al., 2020*). While hsCRP has been widely used as a measure of chronic inflammation, primarily in adults, it is an acute

phase reactant that increases rapidly following acute stimulus and returns to baseline levels within a matter of days and is therefore mostly used as a diagnostic adjunct in children with acute infection or inflammation (*Gabay and Kushner, 1999*). In contrast, there is evidence that GlycA can remain elevated for up to a decade in young adults (*Ritchie et al., 2015*), and it is considered a superior marker of cumulative inflammation burden, though data of GlycA in early life are sparser. Several studies in adults have reported that these two markers are only moderately correlated (*Akinkuolie et al., 2014*; *Gruppen et al., 2015b*), and it is suggested that these markers reflect different (albeit overlapping) inflammatory processes. This is consistent with our findings of distinct metabolomic and lipidomic profiles for these two markers and reflects other findings that show different relationships of GlycA and hsCRP with cardiovascular and metabolic phenotypes. For example, GlycA is independently associated with risk of CVD and with enzymatic esterification of free cholesterol, even after adjustment for hsCRP (*Duprez et al., 2016*; *Gruppen et al., 2015a*; *Muhlestein et al., 2018*). Associations between GlycA and lipolysis rates (*Levine et al., 2020*) and gut microbiome diversity (*Mokkala et al., 2020*) are also stronger than those reported for hsCRP.

## Strengths and limitations

This large prospective cohort study is the first to examine the relationship between infection, inflammation, and plasma NMR metabolomic and LC/MS lipidomic profiles in early life, with implications for later CVD risk. The associations were consistent using either parental-reported infections (potentially prone to reporting bias) or 12-month GlycA as a measure of cumulative inflammatory burden. We found no evidence that birth plasma metabolomic or lipidomic profiles were associated with infection burden in the first 6 months of life (i.e. reverse causation; *Supplementary file 3A and C*). There was some limited evidence that GlycA at 6 months may be associated with subsequent number of infections (*Supplementary file 3B*), but we were able to adjust for this in mediation models to avoid exposure-induced mediator-outcome confounding (*Vanderweele and Vansteelandt, 2009*). The use of metabolomics and lipidomic data from three early life time points (birth, 6 months, and 12 months) is unique and allowed longitudinal analyses to be performed.

Limitations include the use of cross-sectional data for mediation analyses (i.e. 12-month inflammatory markers and 12-month metabolomics/lipidomics), which we addressed in part with mediation models adjusted for infections from birth to 6 months of age and 6-month inflammation. Moreover, GlycA is a marker of cumulative inflammation (*Collier et al., 2019*; *Ritchie et al., 2015*) and is believed to reflect inflammatory events occurring prior to the 12-month time point. Evidence from randomised controlled trials in adults support a causal role of inflammation in CVD risk (*Chunfeng et al., 2021*; *Ridker et al., 2017*), however this study in infants is observational. While we have used a casual framework for mediation analyses, our findings do not demonstrate causality. Despite rigorous adjustment and sensitivity analyses, there may be unmeasured confounding, such as from environmental factors that may modify changes in metabolomic/lipidomic measures following infection. We considered total parent-reported infections, as defining clinical categories of infection given the non-specific symptoms and signs in infancy is challenging. Validation of all parent-reported infections was not feasible and only a minority (70 infections in 555 infants) resulted in medical attention (*Rowland et al., 2021*). However, the strong correlation between metabolomic and lipidomic differences for number of reported infections and for GlycA suggests that parent report is a reasonable measure of infection-induced inflammatory burden. Most childhood infections are viral, and microbial testing is impractical in non-hospitalised children and cannot necessarily differentiate between colonisation and infection. Notwithstanding, distinct lipid differences have been reported for children with bacterial versus viral infections (*Wang et al., 2019*), and different infectious aetiologies in adults have been linked to differential CVD risk (*Cowan et al., 2020*). Finally, the relative lack of racial/ethnic diversity in our cohort may limit generalisability of our findings.

## Conclusions

In summary, we present evidence for higher infection burden in early life leading to proatherogenic and prodiabetic plasma metabolome and lipidome at 12 months of age, and for inflammation partly mediating these relationships. GlycA may be a better marker of early life infection and inflammation burden than hsCRP. These findings suggest that the impact of the cumulative infection and inflammation burdens previously implicated in adult cardiometabolic disease may begin in infancy, thereby

offering opportunities for early prevention. Further work is required to determine the potential consequences these adverse metabolomic profiles in early life have on later risk of disease, and how the relationships between infections, inflammation, and metabolomic and lipidomic profiles might differ across age groups, pathogen type, and clinical severity of infection.

## Data availability

Given the ethics for this study, the individual participant data cannot be made freely available online. Interested parties can access the data used in this study upon reasonable request, with approval by the BIS data custodians. As part of this process, researchers will be required to submit a project concept for approval, to ensure the data is being used responsibly, ethically, and for scientifically sound projects.

Source data files have been uploaded for each of the results figures (*Figures 3–7*) showing the model summary data for all metabolomic and lipidomic measures, including those not presented in figures.

Source code for all analyses have been uploaded as *Source code 1* and *Source code 2*.

## Acknowledgements

The authors thank the BIS participants for the generous contribution they have made to this project. The authors also thank current and past staff for their efforts in recruiting and maintaining the cohort and in obtaining and processing the data and biospecimens.

The members of the BIS Steering Committee are the following: Peter Vuillermin, Anne-Louise Ponsonby, John Carlin, Mimi LK Tang, Fiona Collier, Amy Loughman, Toby Mansell, Lawrence Gray, Martin O'Hely, Richard Saffery, Sarath Ranganathan, David Burgner, Peter Sly, and Leonard Harrison. We thank Terry Dwyer and Katie Allen for their past work as foundation investigators.

## Additional information

### Group author details

**Barwon Infant Study Investigator Group**
**John Carlin**: Murdoch Children's Research Insitute, Royal Children's Hospital, Department of Paediatrics, University of Melbourne, Parkville, Australia; **Amy Loughman**: Institute for Mental and Physical Health and Clinical Translation, School of Medicine, Barwon Health, Deakin University, Geelong, Australia; **Lawrence Gray**: Child Health Research Unit, Barwon Health, Faculty of Health, School of Medicine, Deakin University, Geelong, Australia

### Competing interests

Toby Mansell: TM has received a postdoctoral fellowship from MCRI, is supported by NHMRC funding, has received travel support from MCRI and the University of Melbourne, and received a PhD scholarship from the University of Melbourne. Anne-Louise Ponsonby: ALP is an unpaid scientific advisor for, and has shares in, Dysrupt Labs. ALP has shares in Prevatex Pty Ltd. Mimi LK Tang: MLKT has received funding paid to Murdoch Childen's Research Institute (MCRI) from NHMRC, Prota Theraputics, Abbott Nutrition, the Allergy and Immunology Foundation of Australasia, and the National Children's Research Centre of Ireland, and has received internal research funding from MCRI. MLKT is inventor of 2 patents owned by MCRI relating to allergy treatment and a method to induce tolerance to an allergen. MLKT is a member of the Advisory Boards for Pfizer (has received personal fee) and Anaphylaxis & Anaphylaxis Australia, and of allergy/anaphylaxis-related Committees for the World Allergy Organisation, the International Union of Immunological Societies, the Asia Pacific Association of Allergy Asthma and Clinical Immunology, the American Academy of Allergy Asthma and Immunology, and the Australasian Society of Clinical Immunology and Allergy. MLKT is employee of, and has share options in, Prota Therapeutics. MLKT is an Associate Editor for the Journal of Allergy and Clinical Immunology: Global. Martin O'Hely: MOH has stocks in Prevatex Pty Ltd. Siroon Bekkering: S Bekkering has received postdoctoral grants from the Dutch Heart Foundation and the Dutch Research

Council, and travel support from the European Society for Atherosclerosis. Sarath Ranganathan: SR is Director of the Lung Foundation Australia. SR has stocks/options in Prevatex Pty Ltd. Peter Vuillermin: PV is an inventor on a patent relating to the relationship between maternal carriage of Prevotella. copri and offspring allergic disease, and has stocks/options in Prevatex Pty Ltd. David Burgner: DB has received an Investigator Grant and Project Grant from the Australian National Health and Medical Research Council (NHMRC). Barwon Infant Study Investigator Group: The other authors declare that no competing interests exist.

## Funding

| Funder | Grant reference number | Author |
|---|---|---|
| National Health and Medical Research Council | Senior Research Fellowship 1008396 | Anne-Louise Ponsonby |
| National Health and Medical Research Council | Senior Research Fellowship 1064629 | David Burgner |
| National Health and Medical Research Council | Senior Research Fellowship 1045161 | Richard Saffery |
| National Health and Medical Research Council | Investigator Grant 1110200 | Anne-Louise Ponsonby |
| National Health and Medical Research Council | Investigator Grant 1175744 | David Burgner |
| National Health and Medical Research Council | NHMRC-A*STAR project grant | Peter Meikle |
| Murdoch Children's Research Institute | ECR Fellowship | Toby Mansell |
| Nederlandse Organisatie voor Wetenschappelijk Onderzoek | 452173113 | Siroon Bekkering |

The funders had no role in study design, data collection and interpretation, or the decision to submit the work for publication.

## Author contributions

Toby Mansell, Conceptualization, Data curation, Formal analysis, Methodology, Software, Visualization, Writing – original draft, Writing – review and editing; Richard Saffery, Conceptualization, Funding acquisition, Methodology, Resources, Supervision, Writing – original draft, Writing – review and editing; Satvika Burugupalli, Rebecca Rowland, Data curation, Methodology, Writing – review and editing; Anne-Louise Ponsonby, Funding acquisition, Methodology, Writing – review and editing; Mimi LK Tang, Sarath Ranganathan, Peter D Sly, Peter Vuillermin, Writing – review and editing; Martin O'Hely, Methodology, Writing – review and editing; Siroon Bekkering, Resources, Writing – review and editing; Adam Alexander T Smith, Software, Writing – review and editing; Fiona Collier, Data curation, Resources, Writing – review and editing; Peter Meikle, Resources, Supervision, Writing – original draft, Writing – review and editing; David Burgner, Conceptualization, Funding acquisition, Methodology, Project administration, Resources, Supervision, Writing – original draft, Writing – review and editing

## Author ORCIDs

Toby Mansell ⓘ http://orcid.org/0000-0002-1282-6331
Anne-Louise Ponsonby ⓘ http://orcid.org/0000-0002-6581-3657
Siroon Bekkering ⓘ http://orcid.org/0000-0003-1149-466X
Fiona Collier ⓘ http://orcid.org/0000-0002-5438-480X
Peter Meikle ⓘ http://orcid.org/0000-0002-2593-4665
David Burgner ⓘ http://orcid.org/0000-0002-8304-4302

## Ethics

Human subjects: Participating mothers provided informed consent at recruitment. Ethics approval for this study was granted by the Barwon Health Human Research Ethics Committee (HREC 10/24).

## Decision letter and Author response

Decision letter https://doi.org/10.7554/eLife.75170.sa1

Author response https://doi.org/10.7554/eLife.75170.sa2

## Additional files

### Supplementary files

• Supplementary file 1. Sensitivity analyses. (A) Forest plots summarising sensitivity analyses for models of parent-reported infections from birth to 12 months of age with 12-month nuclear magnetic resonance (NMR) metabolomic measures as outcome. (B) Forest plots summarising sensitivity analyses for models of 12-month glycoprotein acetyls (GlycA) with 12-month NMR metabolomic measures as outcome. (C) Forest plots summarising sensitivity analyses for models of 12-month high-sensitivity C-reactive protein (hsCRP) with 12-month NMR metabolomic measures as outcome. (D) Forest plots summarising sensitivity analyses for models of parent-reported infections from birth to 12 months of age with 12-month liquid chromatography/mass spectrometry (LC/MS) lipidomic measures as outcome. (E) Forest plots summarising sensitivity analyses for models of 12-month GlycA with 12-month LC/MS lipidomic measures as outcome. (F) Forest plots summarising sensitivity analyses for models of 12-month hsCRP with 12-month LC/MS lipidomic measures as outcome.

• Supplementary file 2. Cohort distribution of nuclear magnetic resonance (NMR) metabolomic and liquid chromatography/mass spectrometry (LC/MS) lipidomic measures at 6 and 12 months. (A) Distribution of NMR metabolomic measures in cohort. (B) Distribution of LC/MS lipidomic measures in cohort.

• Supplementary file 3. Forest plots of models investigating reverse causation: models with nuclear magnetic resonance (NMR) metabolomic and liquid chromatography/mass spectrometry (LC/MS) lipidomic measures at birth or 6 months as the exposures and number of parent-reported infections in the following 6 months as the outcome. (A) Difference in log number of parent-reported infections from birth to 6 months of age for each SD increase in log serum NMR metabolomic measure at birth. (B) Difference in log number of parent-reported infections from 6 to 12 months of age for each SD increase in 6-month log plasma NMR metabolomic measure. (C) Difference in log number of parent-reported infections from birth to 6 months of age for each SD increase in log serum LC/MS lipidomic measure at birth. (D) Difference in log number of parent-reported infections from 6 to 12 months of age for each SD increase in 6-month log serum LC/MS lipidomic measure.

• Transparent reporting form

• Source code 1. Code (R Script) for all analyses of nuclear magnetic resonance (NMR) metabolomics data in this study.

• Source code 2. Code (R Script) for all analyses of liquid chromatography/mass spectrometry (LC/MS) lipidomics data in this study.

### Data availability

With the approved ethics for this study, the individual participant data cannot be made freely available online. Interested parties can access the data used in this study upon reasonable request, with approval by the Barwon Infant Study data custodians. As part of this process, researchers will be required to submit a project concept for approval, to ensure the data is being used responsibly, ethically, and for scientifically sound projects. Source data files have been uploaded for each of the results figures (Figures 3 to 7) showing the model summary data for all metabolomic and lipidomic measures, including those not presented in figures. Source code for all analyses have been uploaded as Source Code 1 and 2.

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
