## [Editor Report]

This paper provides data from a population-based cohort study on early life infection and proinflammatory, atherogenic metabolomic and lipidomic profiles at 12 months of age. The authors generated matched infection, metabolomics and lipidomics data from 555 infants in a pre-birth longitudinal cohort and they showed that frequent infant infections are associated with adverse metabolomic and lipidomic profiles. They also report that similar profiles are noted with higher glycoprotein acetyls (GlycA), but not hsCRP. The paper is interesting and assesses the role of infection and markers of inflammation on lipid and metabolic profile of patients. It provides a comprehensive analysis of lipids and metabolites in infants in response to infection.

---

## [Decision Letter]

**Decision letter after peer review:**

Thank you for submitting your article "Early life infection and proinflammatory, atherogenic metabolomic and lipidomic profiles at 12 months of age: a population-based cohort study" for consideration by *eLife*. Your article has been reviewed by 3 peer reviewers, one of whom is a member of our Board of Reviewing Editors, and the evaluation has been overseen by a Senior Editor. The following individual involved in review of your submission has agreed to reveal their identity: Noah Snyder-Mackler (Reviewer #3).

Essential revisions:

1. The paper contains a number of grammatical, spelling and structural errors. The authors are strongly encouraged to review the paper carefully prior to resubmitting the paper.

2. Figures 4 and 5 are difficult to understand. What do each point in the blot represent? What does the line around the data points represent? It is important the authors provide more explanation for readers who do not have familiarity with these plots.

3. The authors need to better define the significance of their findings. For example, although the link between inflammation and changes in metabolomics and lipidomics is known, the authors need to provide a better explanation on why their specific findings are important in the setting of infection in infants.

4. The background and rationale of the introduction should clarify the importance of transient inflammation with infections compared with cumulative burden of inflammation. The authors are suggesting that a higher burden of infections, which are common in the first year of life, are associated with long-lasting inflammation. This does not pan out in terms of hsCRP but does in terms of GlycA but this is not a known robust marker of long-term inflammation. This should be tempered.

5. Are social factors available? For example, a child with two working parents who is placed in daycare is likely to have a higher burden of infections in contrast with one with a stay at home parent? Household income and daycare status would be useful covariates to consider.

6. Are pregnancy-related covariates available? Particularly adjustment for gestational age at delivery, chronic or gestational hypertension, preeclampsia, and/or chronic or gestational diabetes.

7. All results in the text should include measures of imprecision such as 95% CI.

8. Were the children screened and excluded if they had an active infection at the time of measures of inflammatory markers, metabolomics, and lipidomics? How was this assessed?

9. The authors may consider the following reference to align with the AGReMA guidelines for reporting mediation analyses. Lee et al. A Guideline for Reporting Mediation Analyses of Randomized Trials and Observational Studies: The AGReMA Statement. JAMA. 2021;326(11):1045-1056. doi:10.1001/jama.2021.14075; Particularly, a DAG would be useful.

10. Table 2 should include direct effects and clarity on what unit of change for each exposure, mediator, and outcome are the statistics based on? What is a 1-unit change in hsCRP reflect? since these units may vary significantly, use of 1 SD change would be optimal.

11. One technical concern is the variation in the amount of time the blood samples spent between post-processing and storage at -80C (and how they were held during that interval). The authors should be commended for their sensitivity analysis looking at just those samples that were stored for < 4 hours. However, the justification for using only those < 4 hours is not clear. It is also unclear and a bit difficult for a reader to look at Supplementary files (1A and 1B) and draw the authors' conclusion that "this had little difference on the estimated effect sizes observed in analyses with the full cohort". Could the authors make more direct comparisons between the effect sizes to hammer this point home? Given that the sample size is smaller for this analysis, the p-values should be higher (less power), but the effect sizes should be unbiased and relatively similar. Perhaps calculating the correlation between effect sizes would help improve this sensitivity analysis and be good to report in the main text.

12. The authors show that while the correlation between GlycA (or CRP) and number of infections is relatively low (albeit a bit higher for GlycA), the effect sizes of GlycA and infections on the metabolome and lipidome are strongly correlated (and to a lesser extent between CRP and infections). A third comparison here that would be very useful, would be between GlycA and CRP effects on the metabolome and lipidome.

13. The mediation analysis is the least convincing part of the manuscript. It is appreciated that the authors are trying to identify how infections might be translating to adverse metabolomic and lipidomic profiles. However, the justification for GlycA or hsCRP being the mediating mechanisms is not convincing. The authors are implying (statistically through their mediation models) that the effect of the number of infections on many metabolites and lipids is due to the changes in GlycA (or hsCRP). Since both GlycA and number of infections are cumulative measures at 12mo, it is unlikely that one (e.g., GlycA) precedes the other. Rather, as the authors state, GlycA is a marker of cumulative infections. This would preclude it from being a mediator unless the authors have data from earlier timepoints (like previous months) that would provide more support for a mediation effect.

*Reviewer #1 (Recommendations for the authors):*

This is an interesting paper and a large amount of data are provided. I have the following comments:

1. The paper contains a number of grammatical, spelling and structural errors. The authors are strongly encouraged to review the paper carefully prior to resubmitting the paper.

2. Figures 4 and 5 are difficult to understand. What do each point in the blot represent? What does the line around the data points represent? It is important the authors provide more explanation for readers who do not have familiarity with these plots.

3. The authors need to better define the significance of their findings. For example, although the link between inflammation and changes in metabolomics and lipidomics is known, the authors need to provide a better explanation on why their specific findings are important in the setting of infection in infants.

4. Additional environmental variables and their link to changes in childhood serum lipids and metabolome after infection need to be studied and included in the paper.

*Reviewer #2 (Recommendations for the authors):*

Mansell et al. report on associations between infections between birth and 12 months of age and metabolomic/lipidomic profiles. There are several suggestions that may strengthen the study and manuscript:

1) The background and rationale of the introduction should clarify the importance of transient inflammation with infections compared with cumulative burden of inflammation. The authors are suggesting that a higher burden of infections, which are common in the first year of life, are associated with long-lasting inflammation. This does not pan out in terms of hsCRP but does in terms of GlycA but this is not a known robust marker of long-term inflammation. This should be tempered.

2) The introduction suggests that inflammation may be a causal target but this analysis can only identify potential markers. This should be tempered to distinctly argue against any causal findings in this observational analysis?

3) Are social factors available? For example, a child with two working parents who is placed in daycare is likely to have a higher burden of infections in contrast with one with a stay at home parent? Household income and daycare status would be useful covariates to consider.

4) Are pregnancy-related covariates available? Particularly adjustment for gestational age at delivery, chronic or gestational hypertension, preeclampsia, and/or chronic or gestational diabetes.

5) Are higher order gestations (e.g., twins, triplets) excluded from the analysis? If not, a sensitivity analysis excluding them should be considered due to confounding from relatedness and shared maternal characteristics in models

6) All results in the text should include measures of imprecision such as 95% CI

7) Were the children screened and excluded if they had an active infection at the time of measures of inflammatory markers, metabolomics, and lipidomics? How was this assessed?

8) The authors may consider the following reference to align with the AGReMA guidelines for reporting mediation analyses. Lee et al. A Guideline for Reporting Mediation Analyses of Randomized Trials and Observational Studies: The AGReMA Statement. JAMA. 2021;326(11):1045-1056. doi:10.1001/jama.2021.14075; Particularly, a DAG would be useful.

9) The language that GlycA outperforms hsCRP is unclear how this was demonstrated. The authors should provide direct comparisons and supportive data or remove these statements.

10) Table 2 should include direct effects and clarity on what unit of change for each exposure, mediator, and outcome are the statistics based on? What is a 1-unit change in hsCRP reflect? since these units may vary significantly, use of 1 SD change would be optimal

11) since metabolomics and lipidomics were measured at birth, adjustment for these variables in the mediation analysis may be helpful to mitigate concerns of residual confounding

*Reviewer #3 (Recommendations for the authors):*

1. I think the authors should take care in avoiding technical jargon and/or explaining it before using it. For instance, there are multiple points where acronyms are introduced without being first defined, like NMR, TGs, and TG-Os (which I think are nuclear magnetic resonance, and two types of triglycerides). Additionally, technical modifications like lines 211-213 are difficult to parse. What does "used medronic acid to passivate… to avoid peak tailing.." mean to a non-expert?

2. L229: I understand that 1 is added so that you can log-transform any zero values, but is 1 a good value for all metabolites and lipids? Could this offset be scaled to each individual metabolite and lipid based on their lowest non-zero value, such that it doesn't affect the results as much? Additionally, what proportion of the metabolites/lipids measures had 0 values in the samples (and how many 0 values)? In other words: how big of an issue could this +1 offset be?

3. The reverse-causality analysis is really interesting. It would be great if the authors had intermediate time-points (6mo?) rather than just birth to see if there could be some link between inflammatory levels at 6mo (controlling for infection burden) and later life levels.

4. The use of the forest plots is much appreciated, but the effect sizes and CIs are overlapping, so it's really hard to interpret. Especially the lipidomic figures (Figures4 and 5). Here, it might be best to just show the "class totals" and not every species underlying those classes.

The mediation analysis results could also be presented as a forest plot instead of a table. That would make interpretation much easier for the reader.

---

## [Author Response]

Essential revisions:S1. The paper contains a number of grammatical, spelling and structural errors. The authors are strongly encouraged to review the paper carefully prior to resubmitting the paper.

Thank you for flagging this as an issue. There were some errors introduced during late-stage edits. The revised manuscript has been carefully reviewed by several authors for errors.

S2. Figures 4 and 5 are difficult to understand. What do each point in the blot represent? What does the line around the data points represent? It is important the authors provide more explanation for readers who do not have familiarity with these plots.

These points and lines were described in the legend of these figures. E.g., from the old Figure 4 (now Figure 5): ‘Error bars are 95% confidence intervals. In (a) and (b), black points represent class totals, closed dark grey points represent the top 10 species differences by p-value, and hollow points with a black outline represent other species with adjusted p<0.05.’ In response to this comment and the suggestion by Reviewer #3 (see comment R3.4 below), these forest plots have now been simplified; we now only show the lipid class totals in the main figures, instead presenting the forest plot with individual lipid species as a figure supplement. We have also updated these figures to indicate whether a difference in lipid class total has adjusted p-value<0.05 by using closed points, consistent with the NMR metabolomics figures (Figures 3 and 4). For the figure supplement forest plots showing lipid species, we now do not highlight the top 10 species associations by p-value and now do not show the confidence intervals for individual species. We have also revised the description in the figure legends for clarity.

Eg, now for Figure 5: ‘In (a) and (b), error bars are 95% confidence intervals. Closed points represent adjusted p-value<0.05. All models were adjusted for infant age, sex, gestational age, birth weight, maternal household income, smoking during pregnancy, breastfeeding status, and sample processing time. Forest plots depicting individual lipid species within each group are shown in Figure 5—figure supplement 1.’

S3. The authors need to better define the significance of their findings. For example, although the link between inflammation and changes in metabolomics and lipidomics is known, the authors need to provide a better explanation on why their specific findings are important in the setting of infection in infants.

The evolving COVID-19 pandemic has brought the relationship between acute infection and cardiometabolic risk into sharp focus. Notwithstanding, data linking metabolomic and lipidomic perturbations following infection are almost exclusively from adults and older children (Alvarez and Ramos, 1986; Bruzzone et al., 2020; Gallin, Kaye, and O'Leary, 1969; Liuba, Persson, Luoma, Ylä-Herttuala, and Pesonen, 2003; Madsen, Varbo, Tybjærg-Hansen, Frikke-Schmidt, and Nordestgaard, 2018). We therefore consider the findings to be novel and of potentially considerable significance for the following reasons: First, the burden of infection falls largely on pre-school children and ours are among the first data to explore the cardiometabolic associations with common infections in this age group. Second, cardiometabolic risk accrues across the entire life course, as recognised by the American Heart Association who have stated that “*with primordial and primary prevention, CVD is largely preventable*” and that risk stratification and intervention are more likely to be successful if done earlier in life (Weintraub et al., 2011); therefore understanding associations in early life in general populations is an important step towards risk-stratification and targeted interventions. Third, there are few data investigating mass-spec lipidomics in response to common infections in high income countries; indeed, some of the only analogous studies investigated lipidomics following Ebola (Kyle et al., 2019) or acute lower respiratory tract infection (Gao et al., 2019) in adults. Finally, these are the only analyses to explore the potential mediation of these associations by inflammation.

We have therefore added a paragraph to the Discussion that summarises these points (page 7-28, line 510-523):

“The findings are novel and of potentially considerable significance; the burden of infection falls largely on pre-school children and these are among the first data to explore the cardiometabolic associations with common infections in this age group. Second, cardiometabolic risk accrues across the entire life course; the American Heart Association state that “with primordial and primary prevention, CVD is largely preventable” and that risk stratification and intervention are more likely to be successful if done earlier in life (Weintraub et al., 2011) Therefore, understanding associations in early life in general populations is an important step towards risk-stratification and targeted interventions. Third, there are few data investigating LM/CS lipidomics in response to common infections in high income countries; indeed some of the only analogous studies investigated lipidomics following Ebola (Kyle et al., 2019) or acute lower respiratory tract infection (Gao et al., 2019) in adults. These are the only analyses to explore the potential mediation of these associations between infections and omics profiles by inflammation.”

S4. The background and rationale of the introduction should clarify the importance of transient inflammation with infections compared with cumulative burden of inflammation. The authors are suggesting that a higher burden of infections, which are common in the first year of life, are associated with long-lasting inflammation. This does not pan out in terms of hsCRP but does in terms of GlycA but this is not a known robust marker of long-term inflammation. This should be tempered.

We used two markers of inflammation (GlycA and hsCRP). We are well aware that (especially in early childhood) hsCRP is an acute phase reactant and in unlikely to reflect chronic inflammation; in part we included hsCRP as it is so widely used in the adult cardiovascular literature and therefore gives a familiar frame of reference to the use of the more novel marker, GlycA. We acknowledge that GlycA is a suggested rather than proven marker of chronic inflammation, especially in childhood (although we have previously published some evidence from this cohort (Collier et al., 2019)). We have edited the introduction to reflect these points as follows (page 6, line 69-78):

“High-sensitivity C-reactive protein (hsCRP) has been extensively used as a marker of chronic inflammation in adult studies, but is an acute phase reactant in children and may not reflect chronic inflammation in early life. Glycoprotein acetyls (GlycA), is a nuclear magnetic resonance (NMR) composite measure that is suggested to better reflect cumulative, chronic inflammation (Connelly, Otvos, Shalaurova, Playford, and Mehta, 2017). GlycA is an emerging biomarker for cardiometabolic risk (Connelly et al., 2017) that out-performs hsCRP as a predictor of CVD events and mortality (Akinkuolie, Buring, Ridker, and Mora, 2014; Duprez et al., 2016), and of infection-related morbidity and mortality (Ritchie et al., 2015).”

S5. Are social factors available? For example, a child with two working parents who is placed in daycare is likely to have a higher burden of infections in contrast with one with a stay at home parent? Household income and daycare status would be useful covariates to consider.

Annual household income was self-reported by mothers at the 28-weeks gestation time point and included as a covariate in all models. As suggested, child care utilisation may be associated with greater number of infections, but in the opinion of the authors does not seem likely to be itself a confounder of metabolomic/lipidomic outcomes. However, we agree that socio-demographic factors warrant further consideration, we have now performed sensitivity analyses adjusting for two other measures of socioeconomic position and disadvantage available in this cohort instead of household income: (i) *maternal education*, collected by questionnaire at the time of recruitment, and (ii) *the Index of Relative Socio-Economic Disadvantage* (IRSD) from the 2011 Socio-Economic Indexes for Areas (SEIFA) (Pink, 2013), which is a composite of socioeconomic indexes for postcodes within Australia and is based on census data collected every 5 years. With these additional sensitivity analyses, we have now thoroughly considered adjustment for sociodemographic measures capturing different information at a household and neighbourhood level.

These sensitivity analyses are now mentioned in the methods (page 14) and the results (pages 20, 24), and presented as figures in Supplementary Files 1A to 1F. The information on maternal education and SEIFA have been added to the methods (page 9) and Table 1.

Please note that in light of this and other requested analyses (see comments S.6, S.8, R1.4, R2.5), we have restructured the Supplementary Files to present these various sensitivity analyses (including the breastfeeding duration and plasma processing time exclusion sensitivity analyses that were in the previous version of this manuscript) as combined forest plots (Supplementary Files 1A to 1F) allowing for easier visual comparison.

S6. Are pregnancy-related covariates available? Particularly adjustment for gestational age at delivery, chronic or gestational hypertension, preeclampsia, and/or chronic or gestational diabetes.

These covariates (gestational age at delivery, gestational hypertension, preeclampsia, and gestational diabetes) are available. All models have now been adjusted for gestational age. This has attenuated the relationship previously observed between number of infections and the DHA and omega 3 fatty acid measures, and modified some of the associations in the lipidomics data, but the overall findings of the study are otherwise unchanged.

Only small proportions of infants in this cohort with available data were exposed to pre-eclampsia (3.8%) or gestational diabetes (5.5%), so additional adjustment of the primary models for pre-eclampsia and gestational diabetes (and postnatal smoke exposure, see response to R1.4) as now been performed as sensitivity analyses. As gestational hypertension was used to diagnose pre-eclampsia, we have only adjusted for pre-eclampsia and not gestational hypertension.

These sensitivity analyses are now mentioned in the methods (page 14) and the results (pages 20, 24), and presented as figures in Supplementary File 1A to 1F. The information on pre-eclampsia and gestational diabetes data have been added to the methods (page 9) and Table 1.

S7. All results in the text should include measures of imprecision such as 95% CI.

This was originally excluded from the main text (and instead all included in the relevant Source Data) out of concern that it would make the results harder to parse due to the number of results reported. However, following this request, we have now included model estimates and the corresponding 95% CI for all results in the text.

S8. Were the children screened and excluded if they had an active infection at the time of measures of inflammatory markers, metabolomics, and lipidomics? How was this assessed?

Active infection was not an exclusion criterion for participation in this analysis. However, following this comment, we have considered sensitivity analyses excluding infants that may have had an active acute infection based on elevated hsCRP (>5 mg/L as a conservative cut-off indicative of acute infection (Lemiengre et al., 2018; Verbakel et al., 2016)) at 12 months of age.

These sensitivity analyses are now mentioned in the methods (page 14) and the results (pages 20, 24), and presented as figures in Supplementary Files 1A to 1F.

S9. The authors may consider the following reference to align with the AGReMA guidelines for reporting mediation analyses. Lee et al. A Guideline for Reporting Mediation Analyses of Randomized Trials and Observational Studies: The AGReMA Statement. JAMA. 2021;326(11):1045-1056. doi:10.1001/jama.2021.14075; Particularly, a DAG would be useful.

We have now added some revised the mediation analyses and the accompanying text to better align the manuscript with the AGReMA guidelines. This includes:

a) Revising the statistical package used and the estimates of interest. Previously, we used the ‘mediation’ package to estimate average causal direct and indirect effects and calculate percentage mediation from those. However, we felt it would be more appropriate in the context of the AGReMA guidelines and review comment S.10 to instead estimate the natural direct and indirect effects and include those in the results alongside the percentage mediation. For this, we have now used the ‘medflex’ package (Steen, Loeys, Moerkerke, and Vansteelandt, 2017) in R, and have revised the methods text accordingly (page 15-16). Note that we have now calculated percentage mediation based on the natural direct and indirect effects. These percentages are similar (within ~3%) to those using the previous method with average casual effects, and this change in mediation method has not impacted on our overall findings. However, while this revised method has not changed our findings, we feel it is more appropriate with regards to aligning with reporting principles laid out in the AGReMA guidelines.

b) The addition of a representative DAG for the mediation model as a new Figure (Figure 1).

c) Clarification in the methods that a counterfactual-based framework has been used for mediation analyses, and that we are estimating natural direct and indirect effects (page 15, line 297-298).

We have also revised the display item: we have now included a new Figure (Figure 7) to visually display these total and indirect effects as requested in R3.4. What was previously Table 2 has been revised in line with the comment S.10 and is now the Source Data 1 for Figure 7.

S10. Table 2 should include direct effects and clarity on what unit of change for each exposure, mediator, and outcome are the statistics based on? What is a 1-unit change in hsCRP reflect? since these units may vary significantly, use of 1 SD change would be optimal.

As described in the response to comment S.9, we have now added direct and indirect effects to what was previously Table 2 (now Source Data 1 for Figure 7). As in the other analyses in this study, the units are 1 infection change for number of parent-reported infections, and 1 SD change for log-transformed inflammation, metabolomic, and lipidomic measures. This has been included in the figure legend and has now also been added to the footnotes of the Source Data for clarity.

S11. One technical concern is the variation in the amount of time the blood samples spent between post-processing and storage at -80C (and how they were held during that interval). The authors should be commended for their sensitivity analysis looking at just those samples that were stored for < 4 hours. However, the justification for using only those < 4 hours is not clear. It is also unclear and a bit difficult for a reader to look at Supplementary files (1A and 1B) and draw the authors' conclusion that "this had little difference on the estimated effect sizes observed in analyses with the full cohort". Could the authors make more direct comparisons between the effect sizes to hammer this point home? Given that the sample size is smaller for this analysis, the p-values should be higher (less power), but the effect sizes should be unbiased and relatively similar. Perhaps calculating the correlation between effect sizes would help improve this sensitivity analysis and be good to report in the main text.

The <4-hour threshold was selected as a relatively conservative cut-off given the long tail of the sample processing time (see Author response image 1 for a histogram of sample processing time). The majority of samples that were processed within the same day were processed within 4 hours, while samples with >4h processing time were predominantly samples processed the following day after collection. We have now clarified this in the methods section (page 10, lines 170-174). We agree that visually comparing the effect sizes would make the comparison between models easier and have now included a visual comparison for these as a forest plot including this and other supplementary analyses that have been added (e.g., responses to S.6 and S.7) as a supplementary file (Supplementary Files 1A to 1F). We feel that direct comparison of the forest plots may be more readily informative than providing correlation coefficients.

**Author response image 1. sa2fig1:** Distribution of 12-month plasma processing times for the participants in this study.

S12. The authors show that while the correlation between GlycA (or CRP) and number of infections is relatively low (albeit a bit higher for GlycA), the effect sizes of GlycA and infections on the metabolome and lipidome are strongly correlated (and to a lesser extent between CRP and infections). A third comparison here that would be very useful, would be between GlycA and CRP effects on the metabolome and lipidome.

We have now added this comparison between GlycA and hsCRP for both the metabolomic and lipidomic data to the results text (page 21, line 369, and page 24, line 445) and as additional panels in the relevant figures for hsCRP effect sizes (Figures 4C, 6C, and corresponding figure supplements).

S13. The mediation analysis is the least convincing part of the manuscript. It is appreciated that the authors are trying to identify how infections might be translating to adverse metabolomic and lipidomic profiles. However, the justification for GlycA or hsCRP being the mediating mechanisms is not convincing. The authors are implying (statistically through their mediation models) that the effect of the number of infections on many metabolites and lipids is due to the changes in GlycA (or hsCRP). Since both GlycA and number of infections are cumulative measures at 12mo, it is unlikely that one (e.g., GlycA) precedes the other. Rather, as the authors state, GlycA is a marker of cumulative infections. This would preclude it from being a mediator unless the authors have data from earlier timepoints (like previous months) that would provide more support for a mediation effect.

First, we agree that the distinction between infections and GlycA was not made as clearly in the previous version of this manuscript as it should have been. For clarity, we do not consider early life GlycA to solely reflect cumulative infections, but rather consider that early life infections may be a key determinant of early life GlycA, with other non-infectious (and unquantified infectious and non-infectious) exposures also contributing to GlycA levels. Therefore, the intention was not to convey GlycA as a marker solely of cumulative infection burden. We appreciate that describing GlycA in parts of the manuscript as a marker of infection burden did not make this distinction clear, so we have revised these parts to describe GlycA as a marker of inflammation burden instead (e.g., lines 334, 343, 400, 628).

There are few studies that have considered longitudinal changes in GlycA, however studies with longitudinal GlycA measurements in adolescents and adults have reported that recent infection (within 2-3 weeks prior of blood collection) was associated with higher GlycA (Chiesa et al., 2022; Ritchie et al., 2015). There are also several studies investigating COVID-19 that have reported higher GlycA following acute infection compared to non-infected controls or reference populations (Ballout et al., 2021; Bizkarguenaga et al., 2022; Lodge et al., 2021). Together, these studies support the direction of the mediation model we have investigated in this manuscript. However, there is also some evidence in adults that GlycA may predict subsequent risk of infection (Julkunen, Cichońska, Slagboom, Würtz, and Initiative, 2021; Ritchie et al., 2015), and it has been speculated that higher GlycA may reflect a heightened responsiveness of the immune system or immune ‘training’, as we have recently reported from this cohort (Collier et al., 2022).

While originally we did not anticipate having data from an earlier time point than 12 months of age for this study, we very recently received NMR metabolomics data from 6-month plasma, sooner than anticipated, so we have now been able to perform additional analyses utilising these 6-month data. We believe this additional time point greatly strengthens the main findings of this manuscript and in particular addresses the valid queries raised by the reviewers in relation to the causal model.

While some of these additional analyses were not directed requested by the reviewers, we felt it important to include these to strengthen the paper alongside the analyses that were suggested.

Specifically, we have added the following additional analyses to address specific points as follows:

a) We present the additional cross-sectional analysis to provide evidence for a relationship between infection burden from birth to 6 months, 6-month inflammation, and 6-month metabolomic/lipidomic profiles to highlight that some of these associations between infection and omic profiles are evident at 6 months, albeit less so than at 12 months of age.

We therefore provide analysis investigating the relationship between number of parent reported infections from birth to 6 months of age, 6-month inflammation, and 6-month metabolomic/lipidomic measures (methods on page 14, lines 274-276; results on pages 21 and 24, Figure 3—figure supplement 1, Figure 4—figure supplement 1, Figure 5—figure supplement 2, Figure 6—figure supplement 2).

b) To provide evidence that the relationship between infection and adverse omics profiles is less likely to be confounded by unmeasured/unknown early life factors, we further analysed the associations between infections from 6-12 months with 12 month omics, adjusting for the corresponding exposure at 6-months (number of infections from birth to 6 months of age, 6-month GlycA, or 6-month hsCRP) and the corresponding 6-month metabolomic/lipidomic measure (methods on page 14-15, lines 277-280; results on pages 21-22 and 24-25, Figure 3—figure supplement 2, Figure 4—figure supplement 2, Figure 5—figure supplement 3, Figure 6—figure supplement 3).

c) To investigate reverse causality in greater detail, we provide an extension of the previous models of birth metabolomic/lipidomic measures potentially associating with subsequent number of infections. We now have additional data investigating the association of the 6-month metabolomic/lipidomic measurements with number of infections from 6 to 12 months of age, with adjustment for number of infections from birth to 6 months of age (as suggested in comment R3.3) (methods on page 15, lines 283-286; results on pages 22 and 25, Supplementary Files 3B and 3D). Please note that as part of these more in-depth analyses, we have revised the previous reverse causality models with birth metabolomic/lipidomic measures to have infections from birth to 6 months as the outcome instead of infections from birth to 12 months (Supplementary Files 3A and 3C).

d) To strengthen the mediation analyses by adjusting for potential confounding by earlier inflammation, we provide additional mediation analyses investigating 12-month inflammation as a mediator of the effect of number of infections from 6 to 12 months of age on 12-month metabolomic/lipidomic measures, with adjustment for number of infections from birth to 6 months of age, 6-month inflammation, and corresponding 6-month metabolomic/lipidomic measure (to address the concerns raised in this comment and in R2.11) (methods on page 16, results on page 26, Figure 7—figure supplement 1).

Of particular relevance to this specific comment, is evidence from these additional analyses that indicates some limited evidence that 6-month GlycA is modestly associated number of infections from 6 to 12 months of age, after adjustment for number of infections from birth to 6 months of age. This is in keeping with data from adults that GlycA may be a marker of inflammatory potential and predicts long-term risk of severe infection and infection-related mortality (Julkunen et al., 2021; Ritchie et al., 2015). Therefore, we have now included the additional mediation analyses (as mentioned above) investigating 12-month inflammation as a mediator in the relationship between infections from 6-12m and 12-month metabolomic/lipidomic measures, with adjustment for birth-6m infection, 6-month inflammation, and 6-month metabolomic/lipidomic measures. We believe that we have now controlled for the potential of reverse causality of 6-month GlycA on 6m-12m infections and reduced the risk of exposure-induced mediator-outcome confounding and similar confounding. The estimates of proportion mediated in these additional models were similar to those considering infections from birth to 12 months of age.

With these clarifications and additional analyses, we feel these additional analyses strengthen both the rationale and the findings from the mediation analyses.

Reviewer #1 (Recommendations for the authors):This is an interesting paper and a large amount of data are provided. I have the following comments:R1.1. The paper contains a number of grammatical, spelling and structural errors. The authors are strongly encouraged to review the paper carefully prior to resubmitting the paper.

This is S.1 above, please refer to our response there.

R1.2. Figures 4 and 5 are difficult to understand. What do each point in the blot represent? What does the line around the data points represent? It is important the authors provide more explanation for readers who do not have familiarity with these plots.

This is S.2 above, please refer to our response there.

R1.3. The authors need to better define the significance of their findings. For example, although the link between inflammation and changes in metabolomics and lipidomics is known, the authors need to provide a better explanation on why their specific findings are important in the setting of infection in infants.

This is S.3 above, please refer to our response there.

R1.4. Additional environmental variables and their link to changes in childhood serum lipids and metabolome after infection need to be studied and included in the paper.

We have now considered models with additional adjustment for postnatal smoke exposure (methods: page 9, line 147-151, results: Supplementary Files 1A to 1F). We have not adjusted for body weight at 12 months, as effects on 12-month body weight may be in the casual pathway from infections to metabolomic/lipidomic differences. However, all models are adjusted for birth weight z-score. We have also now additionally adjusted all models for gestational age, as well as considered additional adjustment for pre-eclampsia and gestational diabetes (as suggested in S.6) alongside postnatal smoke exposure (2.8% of infants exposed to any postnatal smoke up to 12 months) as sensitivity analysis.

We have also rephrased the comment in the limitations section of the discussion about potential unmeasured confounding to specifically mention environmental factors linked to changes in metabolomic/lipidomic measures after infection (page 31, line 607-609):

“Despite rigorous adjustment and sensitivity analyses, there may be unmeasured confounding, such as from environmental factors that may modify changes in metabolomic/lipidomic measures following infection.”

Reviewer #2 (Recommendations for the authors):Mansell et al. report on associations between infections between birth and 12 months of age and metabolomic/lipidomic profiles. There are several suggestions that may strengthen the study and manuscript:R2.1) The background and rationale of the introduction should clarify the importance of transient inflammation with infections compared with cumulative burden of inflammation. The authors are suggesting that a higher burden of infections, which are common in the first year of life, are associated with long-lasting inflammation. This does not pan out in terms of hsCRP but does in terms of GlycA but this is not a known robust marker of long-term inflammation. This should be tempered.

This is S.4 above, please refer to our response there.

R2.2) The introduction suggests that inflammation may be a causal target but this analysis can only identify potential markers. This should be tempered to distinctly argue against any causal findings in this observational analysis?

There is evidence in adults from a randomised control trial of anti-inflammatory medication (e.g. CANTOS, the Canakinumab Anti-Inflammatory Thrombosis Outcomes Study) (Ridker et al., 2017) and trials of colchicine (Chunfeng, Ping, Yun, Di, and Qi, 2021) that supports inflammation as a causal target to reduce risk of cardiovascular disease. While we had been careful to avoid suggesting that the findings from our study in infants were causal findings, we have now added some text in the Discussion to make this distinction more explicit (page 31, line 602-606):

“Evidence from randomised control trials in adults support a causal role of inflammation in cardiovascular disease risk (Chunfeng et al., 2021; Ridker et al., 2017), however this study in infants is observational. While we have used a casual framework for mediation analyses, our findings do not demonstrate causality.”

R2.3) Are social factors available? For example, a child with two working parents who is placed in daycare is likely to have a higher burden of infections in contrast with one with a stay at home parent? Household income and daycare status would be useful covariates to consider.

This is S.5 above, please refer to our response there.

R2.4) Are pregnancy-related covariates available? Particularly adjustment for gestational age at delivery, chronic or gestational hypertension, preeclampsia, and/or chronic or gestational diabetes.

This is S.6 above, please refer to our response there.

R2.5) Are higher order gestations (e.g., twins, triplets) excluded from the analysis? If not, a sensitivity analysis excluding them should be considered due to confounding from relatedness and shared maternal characteristics in models

Twins were not excluded from the model, but we have now added sensitivity analyses excluding the 5 twins (2 twin pairs, and 1 infant who was a twin to sibling not included in this study due to a lack of 12-month data) from the primary model (methods on page 14, results figures in Supplementary File 1A to 1F).

R2.6) All results in the text should include measures of imprecision such as 95% CI

This is S.7 above, please refer to our response there.

R2.7) Were the children screened and excluded if they had an active infection at the time of measures of inflammatory markers, metabolomics, and lipidomics? How was this assessed?

This is S.8 above, please refer to our response there.

R2.8) The authors may consider the following reference to align with the AGReMA guidelines for reporting mediation analyses. Lee et al. A Guideline for Reporting Mediation Analyses of Randomized Trials and Observational Studies: The AGReMA Statement. JAMA. 2021;326(11):1045-1056. doi:10.1001/jama.2021.14075; Particularly, a DAG would be useful.

This is S.9 above, please refer to our response there.

R2.9) The language that GlycA outperforms hsCRP is unclear how this was demonstrated. The authors should provide direct comparisons and supportive data or remove these statements.

There are data from adults, as cited in the introduction, that GlycA improves risk stratification for CVD, even after adjusting for hsCRP. We have added a sentence to the introduction that briefly elaborates on this statement in relation to CVD risk (page 6, lines 78-82):

“For example, in the Multi-Ethnic Study of Atherosclerosis (n=6523), higher GlycA was associated with increased risk of incidence CVD and death, after adjustment for hsCRP and other inflammatory markers. Conversely, prediction of these outcomes by hsCRP attenuated to null after mutual adjustment (Duprez et al., 2016).”

R2.10) Table 2 should include direct effects and clarity on what unit of change for each exposure, mediator, and outcome are the statistics based on? What is a 1-unit change in hsCRP reflect? since these units may vary significantly, use of 1 SD change would be optimal

This is S.10 above, please refer to our response there.

R2.11) since metabolomics and lipidomics were measured at birth, adjustment for these variables in the mediation analysis may be helpful to mitigate concerns of residual confounding

We have now performed adjusted mediation analysis (discussed in more detail in the response to S.13 above) by assessing mediation by 12-month inflammation in the effect of number of infections from 6 months to 12 months on 12-month metabolomic/lipidomic measures, with adjustment for number of infection from birth to 6 months of age, 6-month inflammation, and 6-month metabolomic/lipidomic measure. We believe this adjusted mediation analysis addresses this concern of residual confounding.

Reviewer #3 (Recommendations for the authors):R3.1. I think the authors should take care in avoiding technical jargon and/or explaining it before using it. For instance, there are multiple points where acronyms are introduced without being first defined, like NMR, TGs, and TG-Os (which I think are nuclear magnetic resonance, and two types of triglycerides). Additionally, technical modifications like lines 211-213 are difficult to parse. What does "used medronic acid to passivate… to avoid peak tailing.." mean to a non-expert?

We have now spelled out all acronyms in full at first use. We have also revised the lipidomics Results sections and figures to include the lipid class names in full instead of predominantly using acronyms for lipid classes.

We appreciate that the omics methodological section is relatively technical; we have attempted to strike a balance being concise and providing sufficient methodological details to reassure those with relevant laboratory expertise that the methods were robust. We would therefore prefer to leave this text unchanged if the editors agree.

R3.2. L229: I understand that 1 is added so that you can log-transform any zero values, but is 1 a good value for all metabolites and lipids? Could this offset be scaled to each individual metabolite and lipid based on their lowest non-zero value, such that it doesn't affect the results as much? Additionally, what proportion of the metabolites/lipids measures had 0 values in the samples (and how many 0 values)? In other words: how big of an issue could this +1 offset be?

Only a small number of metabolomic/lipidomic measures had any 0 values, and most of these measures had only a small number of participants with 0 values. We have added a column for each measure in Supplementary Files 2A and 2B stating the number of 0 values for that measure.

Given that the magnitude of the metabolomic/lipidomic measures differs substantially, we have now redone the analysis as suggested, by offsetting each measure by its minimum non-zero value prior to log-transformation, instead of offsetting all measures by 1. We have also added a column to the tables in Supplementary Files 2A and 2B stating this minimum non-zero measure for each measure.

R3.3. The reverse-causality analysis is really interesting. It would be great if the authors had intermediate time-points (6mo?) rather than just birth to see if there could be some link between inflammatory levels at 6mo (controlling for infection burden) and later life levels.

As described in the response to S.13, we have recently received corresponding metabolomic data for the 6-month time point. As such, we have been able to update the reverse causality modelling: now, we consider the association between birth metabolomic/lipidomic measures and number of infections from birth to 6m, and the association between 6-month metabolomic/lipidomic measures and number of infections from 6m to 12m with adjustment for number of infections from birth to 6m.

R3.4. The use of the forest plots is much appreciated, but the effect sizes and CIs are overlapping, so it's really hard to interpret. Especially the lipidomic figures (Figures4 and 5). Here, it might be best to just show the "class totals" and not every species underlying those classes.The mediation analysis results could also be presented as a forest plot instead of a table. That would make interpretation much easier for the reader.

We have made some revisions to the figures to make them easier to read. Specifically:

a) We have reduced the scale of the x-axis and also increased the width of the NMR metabolomics forest plots for GlycA and hsCRP to avoid the issue of the effect size and CIs overlapping.

b) We have simplified the main LC/MS lipdiomics forest plots to now only show the lipid class totals and not the individual lipid species. We have also now indicated adjusted p>0.05 with an open point and adjusted p<0.05 with a closed point for the lipid class totals to be consistent with the NMR forest plots. We have included simplified versions of the forest plot showing the individuals species as figure supplements: these simplified versions now do not highlight the top 10 species by p-value and also do not show the CIs for the individual lipid species. We have also modified the shades for the individual species to make them less obtrusive on the class total effect sizes and CIs.

As requested, we have also now included a new figure (Figure 7) showing the indirect and total effects of infection for each of the metabolomic/lipidomic measures investigated in the mediation analyses. The Source Data for this figure includes the direct effect as well.

References

Akinkuolie, A. O., Buring, J. E., Ridker, P. M., and Mora, S. (2014). A novel protein glycan biomarker and future cardiovascular disease events. *Journal of the American Heart Association, 3*(5), e001221.

Alvarez, C., and Ramos, A. (1986). Lipids, lipoproteins, and apoproteins in serum during infection. *Clinical Chemistry, 32*(1), 142-145.

Ballout, R. A., Kong, H., Sampson, M., Otvos, J. D., Cox, A. L., Agbor-Enoh, S., and Remaley, A. T. (2021). The NIH Lipo-COVID Study: A pilot NMR investigation of lipoprotein subfractions and other metabolites in patients with severe COVID-19. *Biomedicines, 9*(9), 1090.

Bizkarguenaga, M., Bruzzone, C., Gil‐Redondo, R., SanJuan, I., Martin‐Ruiz, I., Barriales, D.,... Laín, A. (2022). Uneven metabolic and lipidomic profiles in recovered COVID‐19 patients as investigated by plasma NMR metabolomics. *NMR in Biomedicine, 35*(2), e4637.

Bruzzone, C., Bizkarguenaga, M., Gil-Redondo, R., Diercks, T., Arana, E., de Vicuña, A. G.,... San Juan, I. (2020). SARS-CoV-2 infection dysregulates the metabolomic and lipidomic profiles of serum. *Iscience, 23*(10), 101645.

Chiesa, S. T., Charakida, M., Georgiopoulos, G., Roberts, J. D., Stafford, S. J., Park, C.,... Ala‐Korpela, M. (2022). Glycoprotein Acetyls: A Novel Inflammatory Biomarker of Early Cardiovascular Risk in the Young. *Journal of the American Heart Association*, e024380.

Chunfeng, L., Ping, L., Yun, Z., Di, L., and Qi, W. (2021). Colchicine for Coronary Heart Disease: A Meta-Analysis of Randomized Controlled Trials. *Heart Surg Forum, 24*(5), E863-e867. doi:10.1532/hsf.3609

Collier, F., Chau, C., Mansell, T., Faye-Chauhan, K., Vuillermin, P., Ponsonby, A.-L.,... Burgner, D. (2022). Innate Immune Activation and Circulating Inflammatory Markers in Preschool Children. *Frontiers in Immunology, 12*. doi:10.3389/fimmu.2021.830049

Collier, F., Ellul, S., Juonala, M., Ponsonby, A.-L., Vuillermin, P., Saffery, R.,... on behalf of the Barwon Infant Study Investigator, G. (2019). Glycoprotein acetyls (GlycA) at 12 months are associated with high-sensitivity C-reactive protein and early life inflammatory immune measures. *Pediatr Res, 85*(5), 584-585. doi:10.1038/s41390-019-0307-x

Connelly, M. A., Otvos, J. D., Shalaurova, I., Playford, M. P., and Mehta, N. N. (2017). GlycA, a novel biomarker of systemic inflammation and cardiovascular disease risk. *Journal of translational medicine, 15*(1), 219.

Duprez, D. A., Otvos, J., Sanchez, O. A., Mackey, R. H., Tracy, R., and Jacobs Jr, D. R. (2016). Comparison of the predictive value of GlycA and other biomarkers of inflammation for total death, incident cardiovascular events, noncardiovascular and noncancer inflammatory-related events, and total cancer events. *Clinical Chemistry, 62*(7), 1020-1031.

Gallin, J. I., Kaye, D., and O'Leary, W. M. (1969). Serum lipids in infection. *New England Journal of Medicine, 281*(20), 1081-1086.

Gao, D., Zhang, L., Song, D., Lv, J., Wang, L., Zhou, S.,... Zhang, J. (2019). Values of integration between lipidomics and clinical phenomes in patients with acute lung infection, pulmonary embolism, or acute exacerbation of chronic pulmonary diseases: a preliminary study. *Journal of translational medicine, 17*(1), 1-20.

Julkunen, H., Cichońska, A., Slagboom, P. E., Würtz, P., and Initiative, N. H. U. B. (2021). Metabolic biomarker profiling for identification of susceptibility to severe pneumonia and COVID-19 in the general population. *eLife, 10*, e63033.

Kyle, J., Burnum-Johnson, K., Wendler, J., Eisfeld, A., Halfmann, P. J., Watanabe, T.,... Waters, K. (2019). Plasma lipidome reveals critical illness and recovery from human Ebola virus disease. *Proceedings of the National Academy of Sciences, 116*(9), 3919-3928.

Lemiengre, M. B., Verbakel, J. Y., Colman, R., Van Roy, K., De Burghgraeve, T., Buntinx, F.,... De Sutter, A. (2018). Point-of-care CRP matters: normal CRP levels reduce immediate antibiotic prescribing for acutely ill children in primary care: a cluster randomized controlled trial. *Scandinavian journal of primary health care, 36*(4), 423-436.

Liuba, P., Persson, J., Luoma, J., Ylä-Herttuala, S., and Pesonen, E. (2003). Acute infections in children are accompanied by oxidative modification of LDL and decrease of HDL cholesterol, and are followed by thickening of carotid intima–media. *European heart journal, 24*(6), 515-521.

Lodge, S., Nitschke, P., Kimhofer, T., Wist, J., Bong, S.-H., Loo, R. L.,... Lindon, J. C. (2021). Diffusion and relaxation edited proton NMR spectroscopy of plasma reveals a high-fidelity supramolecular biomarker signature of SARS-CoV-2 infection. *Analytical chemistry, 93*(8), 3976-3986.

Madsen, C. M., Varbo, A., Tybjærg-Hansen, A., Frikke-Schmidt, R., and Nordestgaard, B. G. (2018). U-shaped relationship of HDL and risk of infectious disease: two prospective population-based cohort studies. *European heart journal, 39*(14), 1181-1190.

Pink, B. (2013). Socio-economic indexes for areas (SEIFA) 2011. *Canberra: Australian Bureau of Statistics*.

Ridker, P. M., Everett, B. M., Thuren, T., MacFadyen, J. G., Chang, W. H., Ballantyne, C.,... Anker, S. D. (2017). Antiinflammatory therapy with canakinumab for atherosclerotic disease. *New England Journal of Medicine, 377*(12), 1119-1131.

Ritchie, S. C., Würtz, P., Nath, A. P., Abraham, G., Havulinna, A. S., Fearnley, L. G.,... Aalto, K. (2015). The biomarker GlycA is associated with chronic inflammation and predicts long-term risk of severe infection. *Cell Systems, 1*(4), 293-301.

Steen, J., Loeys, T., Moerkerke, B., and Vansteelandt, S. (2017). Medflex: an R package for flexible mediation analysis using natural effect models. *Journal of statistical software, 76*, 1-46.

Verbakel, J. Y., Lemiengre, M. B., De Burghgraeve, T., De Sutter, A., Aertgeerts, B., Shinkins, B.,... Buntinx, F. (2016). Should all acutely ill children in primary care be tested with point-of-care CRP: a cluster randomised trial. *BMC Med, 14*(1), 1-7.

Weintraub, W. S., Daniels, S. R., Burke, L. E., Franklin, B. A., Goff, D. C., Jr., Hayman, L. L.,... Whitsel, L. P. (2011). Value of primordial and primary prevention for cardiovascular disease: a policy statement from the American Heart Association. *Circulation, 124*(8), 967-990. doi:10.1161/CIR.0b013e3182285a81